

# Simultaneous confidence intervals for all pairwise differences between the coefficients of variation of rainfall series in Thailand

Noppadon Yosboonruang, Sa-Aat Niwitpong and Suparat Niwitpong

Department of Applied Statistics, King Mongkut's University of Technology North Bangkok, Bangkok, Thailand

## ABSTRACT

The delta-lognormal distribution is a combination of binomial and lognormal distributions, and so rainfall series that include zero and positive values conform to this distribution. The coefficient of variation is a good tool for measuring the dispersion of rainfall. Statistical estimation can be used not only to illustrate the dispersion of rainfall but also to describe the differences between rainfall dispersions from several areas simultaneously. Therefore, the purpose of this study is to construct simultaneous confidence intervals for all pairwise differences between the coefficients of variation of delta-lognormal distributions using three methods: fiducial generalized confidence interval, Bayesian, and the method of variance estimates recovery. Their performances were gauged by measuring their coverage probabilities together with their expected lengths via Monte Carlo simulation. The results indicate that the Bayesian credible interval using the Jeffreys' rule prior outperformed the others in virtually all cases. Rainfall series from five regions in Thailand were used to demonstrate the efficacies of the proposed methods.

# INTRODUCTION

Thailand is located in Southeast Asia and is classed as a tropical area. It is influenced by both the southwest and northeast monsoons. The southwest monsoon crosses Thailand between mid-May to mid-October (the rainy season) and brings moist air from the Indian Ocean that causes clouds and heavy rain. The northeast monsoon crosses Thailand from mid-October to mid-February (the winter season) causing cold and dry weather. Moreover, the South receives additional heavy rainfall due to moisture coming in from the Gulf of Thailand. The season changes from mid-February to mid-May (the summer season) due to uncertainty in the weather and is influenced by tropical cyclones in the South China Sea, and thus, the weather is generally hot and dry but often with heavy rain and thunderstorms (*Thai Meteorological Department, 2015*). Thailand often endures flooding due to thunderstorms, which can take lives and damage property, especially on farms due to Thailand being an agricultural country. Thailand is divided into five regions according to its

Corresponding author
Suparat Niwitpong,
suparat.n@sci.kmutnb.ac.th

**Table 1  The provinces of each regions in Thailand.**

| Regions | Provinces |
|---|---|
| Northern | Chiang Rai, Mae Hong Son, Chiang Mai, Phayao, Lamphun, Lampang, Phrae, Nan, Uttaradit, Phitsanulok, Sukhothai, Tak, Phichit, Kamphaeng Phet, Phetchabun |
| Northeastern | Nong Khai, Bueng Kan, Loei, Udon Thani,Nong Bua Lam Phu, Nakhon Phanom, Sakon Nakhon, Mukdahan, Khon Kaen, Kalasin, Maha Sarakham, Roi Et, Chaiyaphum, Yasothon, Amnat Charoen, Ubon Ratchathani, Sri Sa Ket, Nakhon Ratchasima, Buri Ram, Surin |
| Central | Nakhon Sawan, Uthai Thani, Chai Nat, Sing Buri, Lop Buri, Ang Thong, Sara buri, Suphan Buri, Ayutthaya, Pathum Thani, Kanchanaburi, Ratchaburi, Nakhon Pathom, Nonthaburi, Bangkok Metropolis, Samut Prakan, Samut Sakhon, Samut Songkhram |
| Eastern | Nakhon Nayok, Prachin Buri, Sra Kaeo, Chachoeng Sao, Chon Buri, Rayong, Chanthaburi, Trat |
| Southern | Phetchaburi, Prachuap Khiri Khan, Chumphon, Surat Thani, Nakhon Si Thammarat, Phatthalung, Songkhla, Pattani, Yala, Narathiwat, Ranong, Phang Nga, Krabi, Phuket Trang, Satun |

climate pattern and meteorological conditions (Table 1) (*Thai Meteorological Department, 2015*). Therefore, it is important to investigate rainfall dispersion in each area to gain preliminary information for formulating policies to mitigate such incidents.

There have been numerous studies on rainfall series that have used the delta-lognormal distribution. *Fukuchi (1988)* derived the distribution of correlation coefficients of rainfall rates from two areas as bivariate lognormal (delta-lognormal). *Kedem (1990)* showed that the average rain rate over an area follows a delta-lognormal distribution. *Shimizu (1993)* and *Kong et al. (2012)* presented the maximum likelihood estimation of the parameters of rainfall series containing zeros that followed a bivariate lognormal distribution. Moreover, examples of rainfall series that conform to a delta-lognormal distribution can be founded in various studies by *Maneerat, Niwitpong & Niwitpong (2019a)*; *Maneerat, Niwitpong & Niwitpong (2019b)*; *Maneerat, Niwitpong & Niwitpong (2020a)*; *Maneerat, Niwitpong & Niwitpong (2020b)*; *Yosboonruang, Niwitpong & Niwitpong (2019b)*; *Yosboonruang, Niwitpong & Niwitpong (2020)*, and *Yosboonruang & Niwitpong (2020)*. In addition, a delta-lognormal distribution has been applied in other fields, such as *Ingram Jr. et al. (2010)*; *Owen & DeRouen (1980)*; *Fletcher (2008)*; *Wu & Hsieh (2014)*, and *Zhou & Tu (2000)*. Constructing the confidence intervals for several parameters of a delta-lognormal distribution used in statistical inference has been of interest to many researchers. Confidence intervals for the delta-lognormal mean were proposed by *Owen & DeRouen (1980)*; *Kvanli, Shen & Deng (1998)*; *Zhou & Tu (2000)*; *Tian (2005)*; *Chen & Zhou (2006)*; *Tian & Wu (2006)*; *Fletcher (2008)*; *Li, Zhou & Tian (2013)*; *Wu & Hsieh (2014)*; *Hasan & Krishnamoorthy (2018)*, and *Maneerat, Niwitpong & Niwitpong (2018)*; *Maneerat, Niwitpong & Niwitpong (2019a)*; *Maneerat, Niwitpong & Niwitpong (2019b)*. Furthermore, the confidence intervals for variance and the coefficient of variation (CV) of a delta-lognormal distribution were presented by *Buntao & Niwitpong (2012)*; *Buntao & Niwitpong (2013)*; *Yosboonruang, Niwitpong & Niwitpong (2018)*; *Yosboonruang, Niwitpong*

*& Niwitpong (2019a)*; *Yosboonruang, Niwitpong & Niwitpong (2019b)*; *Yosboonruang, Niwitpong & Niwitpong (2020)*; *Yosboonruang & Niwitpong (2020)*, and *Maneerat, Niwitpong & Niwitpong (2020a)*; *Maneerat, Niwitpong & Niwitpong (2020b)*.

For statistical inference, the CV, the ratio of the standard deviation to the mean, is a good tool for investigating rainfall dispersion. The advantage of using the CV is that it is unitless and thus, is useful for measuring dispersion in data series with different units or drastically different means. Focusing on inferential statistics, the confidence intervals and functions of the CV for several distributions have been presented. *Wong & Wu (2002)* suggested a small-sample asymptotic method for constructing the confidence intervals for the CV of normal and non-normal distributions when the sample size is very small. *Mahmoudvand & Hassani (2009)* proposed two new methods for constructing the confidence intervals for the CV of a normal distribution and compared them with Miller's, Makay's, Vangel's, and Sharma-Krishna's methods; they found that their proposed methods are more appropriate than the others. *Buntao & Niwitpong (2012)* proposed the generalized pivotal approach (GPA) and a closed-form method for variance estimation for the difference between the CVs of lognormal and delta-lognormal distributions; their results show that the GPA is the most suitable. After that, they constructed the confidence intervals for the ratio of the CVs of delta-lognormal distributions using GPA and the method of variance estimates recovery (MOVER) (*Buntao & Niwitpong, 2013*); their results were similar to the confidence intervals for the difference between the CVs. *Wongkhao, Niwitpong & Niwitpong (2015)* presented the generalized confidence interval (GCI) and MOVER to construct the confidence intervals for the ratio of CVs of normal distributions and then compared their methods with the Verrill and Johnson and bootstrapping methods; they found that GCI and MOVER performed better than the others. *Sangnawakij & Niwitpong (2017a)* proposed MOVER, GCI, and the asymptotic confidence interval (ACI) for constructing the confidence interval for the CV and difference between the CVs of two-parameter exponential distributions; their results show that GCI was appropriate for a single CV and ACI worked well for the difference between the CVs. In addition, confidence intervals were extended by *Sangnawakij & Niwitpong (2017b)* based on the score and Wald intervals for the difference between and ratio of CVs of two gamma distributions; their proposed methods performed well in a comparative study. Recently, *Yosboonruang, Niwitpong & Niwitpong (2018)* proposed GCI and a modified Fletcher method to construct the confidence intervals for the CV of a delta-lognormal distribution and found that GCI was the best. Afterward, they introduced the fiducial GCI (FGCI) and MOVER to construct the confidence intervals for the CV of a delta-lognormal distribution (*Yosboonruang, Niwitpong & Niwitpong, 2019a*). Moreover, they compared the confidence intervals based on FGCI and a Bayesian method for the CV of a delta-lognormal distribution (*Yosboonruang, Niwitpong & Niwitpong, 2019b*); their results indicate that the Bayesian method outperformed FGCI. *Yosboonruang & Niwitpong (2020)* constructed confidence intervals using GCI and MOVER based on variance stabilizing transformation, the Wilson score, and Jeffreys' method for the ratio of the CVs of delta-lognormal distributions; their results show that GCI was the most suitable. *Yosboonruang, Niwitpong & Niwitpong (2020)* presented FGCI and a Bayesian

method to construct the confidence interval for the difference of CVs of delta-lognormal distributions; they concluded that the Bayesian method was the most appropriate.

Since dispersion in the precipitation series for different areas can be the same or different, simultaneous estimation of this for multiple areas has been investigated using various distributions and parameters. *Mandel & Betensky (2008)* introduced an algorithm for simultaneous confidence interval (SCI) construction and then compared bootstrapped and normal-based SCIs in which the limits of the bootstrap intervals were smaller than the normal-based intervals. *Donner & Zou (2011)* used a two-step MOVER approach for constructing SCIs for multiple contrasts of binomial proportions; their proposed method was reasonable for small-to-moderate sample sizes. *Abdel-Karim (2015)* considered three methods: FGCI-MOVER, MOVER-MOVER, and simultaneous FGCI to construct SCIs for the ratio of means of lognormal distributions; they reported that the MOVER-MOVER method outperformed the others. *Li, Song & Shi (2015)* suggested parametric bootstrapping to construct SCIs for all pairwise differences between the means of two-parameter exponential distributions. *Thangjai, Niwitpong & Niwitpong (2019)* presented three methods: MOVER, a computational approach, and FGCI to construct SCIs for all of the differences between the CVs of lognormal distributions; their results show that MOVER was the best and the computational approach performed similarly to MOVER when the sample size was large. In addition, *Thangjai & Niwitpong (2020)* used parametric bootstrapping, GCI, and MOVER for SCI construction for all of the differences between CVs in two-parameter exponential distributions; their results indicate that GCI was the most appropriate in most cases, while MOVER was the best for large sample sizes.

As mentioned above, rainfall series data follow a delta-lognormal distribution. Since our focus is on comparing the dispersion of rainfall from five regions in Thailand, the pairwise differences between the CVs of their rainfall data distributions are an interesting topic to study. Although there have been numerous methods published for constructing SCIs for the differences between the parameters of several types of distributions, constructing SCIs for all of the pairwise differences between the CVs of delta-lognormal distributions has not yet been reported. GCI is a general method that is often used to construct confidence intervals, but FGCI is stronger than GCI since it provides asymptotically correct frequentist coverage (*Hannig, Abdel-Karim & Iyer, 2006*). Moreover, previous researchers have reported that MOVER is an appropriate method for constructing the SCIs for various parameters of several types of distributions. Therefore, one of ours aims was to construct SCIs for this scenario based on FGCI and compare them with ones based on MOVER and Bayesian methodology. The coverage probability, the probability that the confidence interval of the estimate covers the value of the parameter, together with the expected length were used to estimate the performance of the confidence intervals.

## METHODS

Let $X_i = (X_{i1}, X_{i2}, \ldots, X_{in_i})$, $i = 1, 2, \ldots, k$ be a random sample from $k$ independent delta-lognormal distributions, denoted by $X_{ij} \sim \Delta(\mu_i, \sigma_i^2, \delta_{i(0)})$, where $\delta_{i(0)} = P(X_{ij} = 0)$. Since this distribution contains zero and positive values, then the zero values follow

a binomial distribution and the positive values a lognormal distribution denoted by $X_{ij} = 0 \sim Bin(n_i, \delta_{i(0)})$ and $Y_{ij} = \ln(X_{ij}) \sim N(\mu_i, \sigma_i^2)$, respectively. Moreover, let $n_{i(0)}$ and $n_{i(1)}$ be the numbers of zero and positive values, respectively, where $n_i = n_{i(0)} + n_{i(1)}$. The distribution function of a delta-lognormal distribution is given by

$$f(x_{ij}; \mu_i, \sigma_i^2, \delta_{i(0)}) = \begin{cases} \delta_{i(0)} & ; x_{ij} = 0 \\ \delta_{i(1)} \dfrac{1}{\sqrt{2\pi}\sigma_i} \left(\dfrac{1}{x_{ij}}\right) \exp\left\{-\dfrac{[\ln(x_{ij}) - \mu_i]^2}{2\sigma_i^2}\right\} & ; x_{ij} > 0, \end{cases} \tag{1}$$

where $\delta_{i(1)} = 1 - \delta_{i(0)}$. Following *Aitchison (1955)*, the respective population mean and variance of $X_i$ are

$$E(X_i) = \mu_{X_i} = \delta_{i(1)} \exp\left(\mu_i + \frac{\sigma_i^2}{2}\right) \tag{2}$$

and

$$Var(X_i) = \sigma_{X_i}^2 = \delta_{i(1)} \exp(2\mu_i + \sigma_i^2)\left[\exp(\sigma_i^2) - \delta_{i(1)}\right]. \tag{3}$$

Following this, the CV of $X_i$ can be expressed as

$$CV(X_i) = \nu_i = \sqrt{\frac{\exp(\sigma_i^2) - \delta_{i(1)}}{\delta_{i(1)}}}. \tag{4}$$

Since we are interested in constructing the SCIs for all pairwise differences between the CVs, then

$$\nu_{il} = \nu_i - \nu_l = \sqrt{\frac{\exp(\sigma_i^2) - \delta_{i(1)}}{\delta_{i(1)}}} - \sqrt{\frac{\exp(\sigma_l^2) - \delta_{l(1)}}{\delta_{l(1)}}}, \tag{5}$$

where $i, l = 1, 2, \ldots, k$ and $i \neq l$. The maximum likelihood estimators of $\delta_{i(1)}$ and $\mu_i$ are $\hat{\delta}_{i(1)} = n_{i(1)}/n_i$ and $\hat{\mu}_i = \sum_{j=1}^{n_{i(1)}} \ln(x_{ij})/n_{i(1)}$, respectively. Furthermore, the unbiased estimator for $\sigma_i^2$ is $\hat{\sigma}_i^2 = \sum_{j=1}^{n_{i(1)}} [\ln(x_{ij}) - \hat{\mu}_i]^2/(n_{i(1)} - 1)$.

Assume that $\hat{\delta}_{i(1)}$ and $\hat{\sigma}_i^2$ are independent, then the maximum likelihood estimator of $\nu_i$ can be defined as

$$\hat{\nu}_i = \sqrt{\frac{\exp(\hat{\sigma}_i^2) - \hat{\delta}_{i(1)}}{\hat{\delta}_{i(1)}}}. \tag{6}$$

Similarly,

$$\hat{\nu}_{il} = \hat{\nu}_i - \hat{\nu}_l = \sqrt{\frac{\exp(\hat{\sigma}_i^2) - \hat{\delta}_{i(1)}}{\hat{\delta}_{i(1)}}} - \sqrt{\frac{\exp(\hat{\sigma}_l^2) - \hat{\delta}_{l(1)}}{\hat{\delta}_{l(1)}}}, \tag{7}$$

where $i, l = 1, 2, \ldots, k$ and $i \neq l$.

According to *Yosboonruang, Niwitpong & Niwitpong (2020)*, the estimated variance of $\hat{v}_i - \hat{v}_l$ can be expressed as

$$\hat{Var}(\hat{v}_i - \hat{v}_l) = \frac{\left\{\left[\ln\left(\hat{\delta}_{i(1)}\right) + \ln\left(\frac{\exp(\hat{\sigma}_i^2) - \hat{\delta}_{i(1)}}{\hat{\delta}_{i(1)}} + 1\right)\right]\left[\frac{\exp(\hat{\sigma}_i^2) - \hat{\delta}_{i(1)}}{\hat{\delta}_{i(1)}} + 1\right]\right\}^2}{2n_i\left[\frac{\exp(\hat{\sigma}_i^2) - \hat{\delta}_{i(1)}}{\hat{\delta}_{i(1)}}\right]}$$

$$+ \frac{\left\{\left[\ln\left(\hat{\delta}_{l(1)}\right) + \ln\left(\frac{\exp(\hat{\sigma}_l^2) - \hat{\delta}_{l(1)}}{\hat{\delta}_{l(1)}} + 1\right)\right]\left[\frac{\exp(\hat{\sigma}_l^2) - \hat{\delta}_{l(1)}}{\hat{\delta}_{l(1)}} + 1\right]\right\}^2}{2n_l\left[\frac{\exp(\hat{\sigma}_l^2) - \hat{\delta}_{l(1)}}{\hat{\delta}_{l(1)}}\right]}, \tag{8}$$

where $i, l = 1, 2, \ldots, k$ and $i \neq l$.

## The simultaneous FGCIs

To construct the simultaneous FGCIs, a fiducial generalized pivotal quantity (FGPQ), which is a subclass of the generalized pivotal quantity (GPQ) (*Hannig, Iyer & Patterson, 2006*), is presented as follows.

**Definition 1** Let $X_i = (X_{i1}, X_{i2}, \ldots, X_{in_i}), i = 1, 2, \ldots, k$ be a random sample from $k$ independent delta-lognormal distributions with parameters of interest $(\sigma_i^2, \delta_{i(1)})$ and nuisance parameter $\mu_i$. Let $x_i = (x_{i1}, x_{i2}, \ldots, x_{in_i}), i = 1, 2, \ldots, k$ be an observed value of $X_i$. GPQ $R(X_i; x_i, \mu_i, \sigma_i^2, \delta_{i(1)})$ is called an FGPQ if it corresponds with the following two conditions (*Weerahandi, 1993*; *Hannig, Iyer & Patterson, 2006*): 1. For a given $x_i$, the conditional distribution of $R(X_i; x_i, \mu_i, \sigma_i^2, \delta_{i(1)})$ is free of $\mu_i$. 2. The observed value of $R(X_i; x_i, \mu_i, \sigma_i^2, \delta_{i(1)})$ at $X_i = x_i, r(x_i; x_i, \mu_i, \sigma_i^2, \delta_{i(1)})$ is the parameter of interest.

The FGPQs for $\sigma_i^2$ and $\delta_{i(1)}$ can be constructed by applying Definition 1. According to *Hannig, Iyer & Patterson (2006)* and *Li, Zhou & Tian (2013)*, the respective FGPQs for $\delta_{i(1)}$ and $\sigma_i^2$ are

$$R_{\delta_{i(1)}} \sim \frac{1}{2}Beta\left(n_{i(1)}, n_{i(0)} + 1\right) + \frac{1}{2}Beta\left(n_{i(1)} + 1, n_{i(0)}\right) \tag{9}$$

and

$$R_{\sigma_i^2} = \frac{(n_{i(1)} - 1)\hat{\sigma}_i^2}{U_i}, \tag{10}$$

where $U_i \sim \chi_{n_{i(1)}-1}^2$. Following this, the FGPQ for $v_i$ is simply

$$R_{v_i} = \sqrt{\frac{\exp\left(R_{\sigma_i^2}\right) - R_{\delta_{i(1)}}}{R_{\delta_{i(1)}}}}. \tag{11}$$

Hence, the FGPQ for the differences between two independent CVs can be expressed as

$$R_{v_{il}} = R_{v_i} - R_{v_l} = \sqrt{\frac{\exp\left(R_{\sigma_i^2}\right) - R_{\delta_{i(1)}}}{R_{\delta_{i(1)}}}} - \sqrt{\frac{\exp\left(R_{\sigma_l^2}\right) - R_{\delta_{l(1)}}}{R_{\delta_{l(1)}}}}, \tag{12}$$

where $i, l = 1, 2, \ldots, k$ and $i \neq l$.

Therefore, the $100(1-\alpha)\%$ two-sided SCI for $v_i - v_l$ based on the FGCI method can be written as $L_{il} \leqslant v_{il} \leqslant U_{il}$, where $L_{il}$ and $U_{il}$ are the $\alpha/2$-th and $(1-\alpha/2)$-th quantiles of $R_{v_{il}}$, respectively.

**Theorem 1** *Let $X_i = (X_{i1}, X_{i2}, \ldots, X_{in_i})$, $i = 1, 2, \ldots, k$ be a random sample from $k$ independent delta-lognormal distributions with mean $\mu_i$, variance $\sigma_i^2$, and probability of zero values $\delta_{i(0)}$. Let $v_i = \sqrt{[\exp(\sigma_i^2) - \delta_{i(1)}]/\delta_{i(1)}}$ and $v_l = \sqrt{[\exp(\sigma_l^2) - \delta_{l1}]/\delta_{l1}}$ for $i, l = 1, 2, \ldots, k$ and $i \neq l$ be the CV of $X_i$ and $X_l$, respectively. Furthermore, let $\hat{v}_i$ and $\hat{v}_l$ be the estimators of $v_i$ and $v_l$, respectively. The estimator for the variance of the difference between $v_i$ and $v_l$ is $\hat{Var}(\hat{v}_i - \hat{v}_l)$. Let $n_i$ be the sample size of the i-th random sample and $n = n_1 + n_2 + \ldots + n_k$. Assume that $n_i/n \to r_i$ as $n \to \infty$ where $0 < r_i < 1$. Therefore,*

$$P\left[R_{v_{il}}(\alpha/2) \leq R_{v_{il}} \leq R_{v_{il}}(1-\alpha/2), \forall i \neq l\right] \to 1 - \alpha. \tag{13}$$

**Proof** Since

$$P\left[R_{v_{il}}(\alpha/2) \leq R_{v_{il}} \leq R_{v_{il}}(1-\alpha/2), \forall i \neq l\right] = P\left[L_{il} \leqslant v_i - v_l \leqslant U_{il}, \forall i \neq l\right],$$

where $[L_{il}, U_{il}] = \hat{v}_i - \hat{v}_l \pm d_{1-\alpha}\sqrt{\hat{Var}(\hat{v}_i - \hat{v}_l)}$ and $d_{1-\alpha}$ denotes the $(1-\alpha)$-th quantile of $R_{v_{il}}$. Thus,

$$P\left[L_{il} \leqslant v_i - v_l \leqslant U_{il}, \forall i \neq l\right] = P\left[\max_{i \neq l}\left|\frac{\hat{v}_i - \hat{v}_l - (v_i - v_l)}{\sqrt{\hat{Var}(\hat{v}_i - \hat{v}_l)}}\right| \leq d_{1-\alpha}\right]$$

$$= P\left[D_n \leq d_{1-\alpha}\right].$$

Accordingly,

$$P\left[L_{il} \leqslant v_i - v_l \leqslant U_{il}, \forall i \neq l\right] \to 1 - \alpha.$$

This implies that

$$P\left[R_{v_{il}}(\alpha/2) \leq R_{v_{il}} \leq R_{v_{il}}(1-\alpha/2), \forall i \neq l\right] \to 1 - \alpha. \quad \square$$

## The Bayesian method

According to the distributions of $X_i$ for $i = 1, 2, \ldots, k$ with the unknown parameters $\mu_i, \sigma_i^2$, and $\delta_{i(0)}$, where $\delta_{i(0)} = 1 - \delta_{i(1)}$, the joint likelihood function of $k$ independent delta-lognormal distributions is

$$L\left(\mu_i, \sigma_i^2, \delta_{i(1)}|x_{ij}\right) \propto \prod_{i=1}^{k} \left(1 - \delta_{i(1)}\right)^{n_{i(0)}} \delta_{i(1)}^{n_{i(1)}} \left(\sigma_i^2\right)^{-\frac{n_{i(1)}}{2}} \exp\left\{-\frac{1}{2\sigma_i^2}\sum_{j=1}^{n_{i(1)}}\left[\ln\left(x_{ij}\right) - \mu_i\right]^2\right\}. \tag{14}$$

By applying the second-order partial derivative of the log-likelihood function with respect to the unknown parameters, the Fisher information matrix of the unknown parameters can be written as

$$I\left(\mu_i, \sigma_i^2, \delta_{i(1)}\right) =$$

$$\text{diag}\left[\frac{n_1}{\left(1-\delta_{1(1)}\right)\delta_{1(1)}} \quad \frac{n_1\delta_{1(1)}}{\sigma_1^2} \quad \frac{n_1\delta_{1(1)}}{2\left(\sigma_1^2\right)^2} \quad \cdots \quad \cdots \quad \cdots \quad \frac{n_k}{\left(1-\delta_{k(1)}\right)\delta_{k(1)}} \quad \frac{n_k\delta_{k(1)}}{\sigma_k^2} \quad \frac{n_k\delta_{k(1)}}{2\left(\sigma_k^2\right)^2}\right].$$
$$(15)$$

In this paper, we constructed both of equal-tailed SCIs based on simulation data and simultaneous credible intervals based on information from a simulation study of their prior distributions using two forms of Bayesian prior; the suitability of the Jeffreys' rule and uniform priors was determined by considering the values of a random variable of their posterior distributions that correspond to those for a delta-lognormal distribution. See also, *Yosboonruang, Niwitpong & Niwitpong (2019b)* and *Yosboonruang, Niwitpong & Niwitpong (2020)*.

### The Jeffreys' rule prior

The Jeffreys' rule prior is obtained from the square root of the determinant of the Fisher information matrix (*Jeffreys, 1946*). It is well known that a delta-lognormal distribution comprises lognormal and binomial distributions. From the CVs in Eq. (4), the parameters of interest are $\sigma_i^2$ and $\delta_{i(1)}$, and the Jeffreys' rule priors for these parameters are $p\left(\sigma_i^2\right) \propto \sigma_i^{-3}$ and $p\left(\delta_{i(1)}\right) \propto \left(1-\delta_{i(1)}\right)^{-\frac{1}{2}}\delta_{i(1)}^{\frac{1}{2}}$, respectively. Assuming that $\sigma_i^2$ and $\delta_{i(1)}$ are independent, the prior distribution for a delta-lognormal distribution can be defined as $p\left(\sigma_i^2, \delta_{i(1)}\right) \propto \sigma_i^{-3}\left(1-\delta_{i(1)}\right)^{-\frac{1}{2}}\delta_{i(1)}^{\frac{1}{2}}$. By combining the likelihood function and the prior distribution of a delta-lognormal distribution, the joint posterior density function can be written as

$$p\left(\sigma_i^2, \delta_{i(1)}|x_{ij}\right) = \prod_{i=1}^{k}\frac{1}{Beta\left(n_{i(0)}+\frac{1}{2}, n_{i(1)}+\frac{3}{2}\right)}\left(1-\delta_{i(1)}\right)^{\left(n_{i(0)}+\frac{1}{2}\right)-1}\delta_{i(1)}^{\left(n_{i(1)}+\frac{3}{2}\right)-1}$$

$$\times \frac{1}{\sqrt{2\pi}\frac{\sigma_i}{\sqrt{n_{i(1)}}}}\exp\left[-\frac{1}{2\frac{\sigma_i^2}{n_{i(1)}}}\left(\mu_i-\hat{\mu}_i\right)^2\right]\frac{\left(\frac{n_{i(1)}\hat{\sigma}_i^2}{2}\right)^{\frac{n_{i(1)}}{2}}}{\Gamma\left(\frac{n_{i(1)}}{2}\right)}\left(\sigma_i^2\right)^{-\frac{n_{i(1)}}{2}-1} \quad (16)$$

$$\times \exp\left(-\frac{\frac{n_{i(1)}\hat{\sigma}_i^2}{2}}{\sigma_i^2}\right),$$

where $\hat{\mu}_i = \sum_{j=1}^{n_{i(1)}}\ln\left(x_{ij}\right)/n_{i(1)}$, and $\hat{\sigma}_i^2 = \sum_{j=1}^{n_{i(1)}}\left[\ln\left(x_{ij}\right)-\hat{\mu}_i\right]^2/\left(n_{i(1)}-1\right)$.

By integrating Eq. (16), the respective posterior distributions of $\sigma_i^2$ and $\delta_{i(1)}$ are derived as

$$p\left(\sigma_i^2|x_{ij}\right) \propto \prod_{i=1}^{k}\frac{\left(\frac{n_{i(1)}\hat{\sigma}_i^2}{2}\right)^{\frac{n_{i(1)}}{2}}}{\Gamma\left(\frac{n_{i(1)}}{2}\right)}\left(\sigma_i^2\right)^{-\frac{n_{i(1)}}{2}-1}\exp\left(-\frac{\frac{n_{i(1)}\hat{\sigma}_i^2}{2}}{\sigma_i^2}\right), \quad (17)$$

and

$$p\left(\delta_{i(1)}|x_{ij}\right) \propto \prod_{i=1}^{k}\frac{1}{Beta\left(n_{i(0)}+\frac{1}{2}, n_{i(1)}+\frac{3}{2}\right)}\left(1-\delta_{i(1)}\right)^{\left(n_{i(0)}+\frac{1}{2}\right)-1}\delta_{i(1)}^{\left(n_{i(1)}+\frac{3}{2}\right)-1}. \quad (18)$$

It should be noted that $p(\sigma_i^2|x_{ij})$ follows an inverse gamma distribution and $p(\delta_{i(1)}|x_{ij})$ follows a beta distribution, denoted by $\sigma_i^2|x_{ij} \sim Inv - Gamma\left(n_{i(1)}/2, n_{i(1)}\acute{\sigma}_i^2/2\right)$ and $\delta_{i(1)}|x_{ij} \sim Beta\left(n_{i(0)}+1/2, \; n_{i(1)}+3/2\right)$, respectively. Consequently, $\sigma_i^2|x_{ij}$ and $\delta_{i(1)}|x_{ij}$ can be substituted into (5) to construct the equal-tailed SCI and the simultaneous credible interval, respectively.

### The uniform prior

Since the uniform prior has a constant function for the prior probability (*Stone, 2013*), then the uniform priors of $\sigma_i^2$ and $\delta_{i(1)}$ are 1, denoted by $p(\sigma_i^2) \propto 1$ and $p(\delta_{i(1)}) \propto 1$, respectively. Afterward, the uniform prior for a delta-lognormal distribution becomes $p(\sigma_i^2, \delta_{i(1)}) \propto 1$. Similar to Eq. (16), the joint posterior density function is obtained by combining $p(\sigma_i^2, \delta_{i(1)})$ with the likelihood function from Eq. (14). Subsequently, we obtain the posterior of $\sigma_i^2$ and $\delta_{i(1)}$ by integrating the joint posterior density function with respect to the others. Thus, the posterior distribution is $\sigma_i^2|x_{ij} \sim Inv - Gamma\left[\left(n_{i(1)}-2\right)/2, \left(n_{i(1)}-2\right)\hat{\sigma}_i^2/2\right]$ for $\sigma_i^2$ and $\delta_{i(1)}|x_{ij} \sim Beta\left(n_{i(0)}+1, n_{i(1)}+1\right)$ for $\delta_{i(1)}$.

Therefore, the $100(1-\alpha)\%$ equal-tailed SCI and simultaneous credible interval for $v_{il}$ based on the Bayesian method are $L_{il} \leq v_{il} \leq U_{il}$, where $L_{il}$ and $U_{il}$ are the lower and upper bounds of the intervals, respectively.

**Theorem 2** *Let* $X_i = (X_{i1}, X_{i2}, \ldots, X_{in_i}) \sim \Delta\left(\mu_i, \sigma_i^2, \delta_{i(1)}\right)$, *where* $i = 1, 2, \ldots, k$ *and* $\delta_{i(0)} = 1 - \delta_{i(1)}$, *with sample sizes* $n_1, n_2, \ldots, n_k$ *and* $n = n_1 + n_2 + \ldots + n_k$. *Let* $r_i = n_i/n$ *as* $n \rightarrow \infty$, *where* $0 < r_i < 1$. *For* $i, l = 1, 2, \ldots, k$ *and* $i \neq l$, *let* $v_i = \sqrt{\left[\exp\left(\sigma_i^2\right) - \delta_{i(1)}\right]/\delta_{i(1)}}$ *and* $v_l = \sqrt{\left[\exp\left(\sigma_l^2\right) - \delta_{l(1)}\right]/\delta_{l(1)}}$ *be the CVs of* $X_i$ *and* $X_l$, *respectively. Let* $\hat{v}_i$ *and* $\hat{v}_l$ *be the estimators of* $v_i$ *and* $v_l$, *respectively. An estimator for the variance of the difference between* $v_i$ *and* $v_l$ *is* $\hat{Var}(\hat{v}_i - \hat{v}_l)$. *Let* $p(\sigma_i^2, \delta_{i(1)})$ *and* $p(\sigma_i^2, \delta_{i(1)}|x_{ij})$ *be the prior distribution and the joint posterior density function for delta-lognormal distribution, respectively. Therefore,*

$$P\left[L_{il} \leq v_{il} \leq U_{il}, \forall i \neq l\right] \rightarrow 1 - \alpha. \quad \square \tag{19}$$

**Proof** The proof is similar to Theorem 1.

> **Algorithm 1:** For the FGCI and Bayesian methods
> Step 1.  Generate random samples $X_i$, $i = 1, 2, \ldots, k$, with sample sizes $n_1, n_2, \ldots, n_k$ and calculate $\hat{\delta}_{i(1)}$ and $\hat{\sigma}_i^2$.
> Step 2.  Generate $U_i \sim \chi_{n_{i(1)}-1}^2, Beta\left(n_{i(1)}, n_{i(1)}+1\right), Beta\left(n_{i(1)}+1, n_{i(1)}\right),$ $Beta\left(n_{i(0)}+1/2, n_{i(1)}+3/2\right), Beta\left(n_{i(0)}+1, n_{i(1)}+1\right), Inv - Gamma\left(n_{i(1)}/2, n_{i(1)}\hat{\sigma}_i^2/2\right),$ and $Inv - Gamma\left[\left(n_{i(1)}-2\right)/2, \left(n_{i(1)}-2\right)\hat{\sigma}_i^2/2\right].$
> Step 3.  Calculate $R_{\delta_{i(1)}}, R_{\sigma_i^2}, R_{v_i}, R_{v_l}, v_i,$ and $v_l$.
> Step 4.  Repeat Steps 2–3 5,000 times.
> Step 5.  Compute the 95% SCIs for $v_{il}$.
> Step 6.  Repeat Steps 1–5 15,000 times.

### MOVER

The concept of MOVER proposed by *Donner & Zou (2012)* can be applied to construct the $100(1-\alpha)\%$ two-sided confidence interval of $v_i - v_l$ for $i, l = 1, 2, \ldots, k$ and $i \neq l$, for

which $L_{il} \leq v_{il} \leq U_{il}$ where $L_{il}$ and $U_{il}$ denote the lower and upper limits of the confidence interval, respectively, expressed as

$$L_{il} = \hat{v}_i - \hat{v}_l - \sqrt{(\hat{v}_i - l_i)^2 + (u_l - \hat{v}_l)^2} \tag{20}$$

and

$$U_{il} = \hat{v}_i - \hat{v}_l + \sqrt{(u_i - \hat{v}_i)^2 + (\hat{v}_l - l_l)^2}, \tag{21}$$

where $i, l = 1, 2, \ldots, k$ and $i \neq l$. From (4), the parameters of interest are $\delta_{i(1)}$ and $\sigma_i^2$, and so the confidence intervals for these parameters can be constructed.

Since the unbiased estimator of $\sigma_i^2$ is given by $\hat{\sigma}_i^2 = \sum_{j=1}^{n_{i(1)}} [\ln(x_{ij}) - \hat{\mu}_i]^2 / (n_{i(1)} - 1)$, for $i = 1, 2, \ldots, k$ and where $(n_{i(1)} - 1)\hat{\sigma}_i^2 / \sigma^2 \sim \chi^2_{n_{i(1)}-1}$. Consequently, the respective lower and upper bounds for $\sigma_i^2$ are defined as

$$l_{\sigma_i^2} = \frac{(n_{i(1)} - 1)\hat{\sigma}_i^2}{\chi^2_{1-\frac{\alpha}{2}, n_{i(1)}-1}} \tag{22}$$

and

$$u_{\sigma_i^2} = \frac{(n_{i(1)} - 1)\hat{\sigma}_i^2}{\chi^2_{\frac{\alpha}{2}, n_{i(1)}-1}}. \tag{23}$$

The score method proposed by *Wilson (1927)* is used to construct the confidence limits for $\delta_{i(1)}$. According to *Brown, Cai & DasGupta (2001)* and *Donner & Zou (2011)*, the respective lower and upper limits of $\delta_{i(1)}$ are given by

$$l_{\delta_{i(1)}} = \frac{n_{i(1)} + \frac{Z_{i(\alpha/2)}^2}{2}}{n_i + Z_{i(\alpha/2)}^2} - Z_{i(\alpha/2)} \frac{\sqrt{\frac{n_{i(0)} n_{i(1)}}{n_i} + \frac{Z_{i(\alpha/2)}^2}{4}}}{n_i + Z_{i(\alpha/2)}^2} \tag{24}$$

and

$$u_{\delta_{i(1)}} = \frac{n_{i(1)} + \frac{Z_{i(\alpha/2)}^2}{2}}{n_i + Z_{i(\alpha/2)}^2} + Z_{i(\alpha/2)} \frac{\sqrt{\frac{n_{i(0)} n_{i(1)}}{n_i} + \frac{Z_{i(\alpha/2)}^2}{4}}}{n_i + Z_{i(\alpha/2)}^2}, \tag{25}$$

where $Z_i, i = 1, 2, \ldots, k$ follow a standard normal distribution. This approach is similar to constructing the confidence limits for $\sigma_l^2$ and $\delta_{l(1)}$.

Therefore, the $100(1-\alpha)\%$ two-sided SCIs for $v_i - v_l$ based on the MOVER method are

$$SCI_{il} = [L_{il}, U_{il}], \tag{26}$$

where $i, l = 1, 2, \ldots, k$ and $i \neq l$.

**Theorem 3** *Let $X_i = (X_{i1}, X_{i2}, \ldots, X_{in_i})$, $i = 1, 2, \ldots, k$, be random samples from $k$ independent delta-lognormal distributions with mean $\mu_i$, variance $\sigma_i^2$, and probability of zero values $\delta_{i(0)}$. Furthermore, let the sample size of the $i$-th random sample be $n_i$, where $n = n_1 + n_2 + \ldots + n_k$ and $r_i = n_i/n$ as $n \to \infty$, for which $0 < r_i < 1$. Let*

$v_i = \sqrt{\left[\exp\left(\sigma_i^2\right) - \delta_{i(1)}\right]/\delta_{i(1)}}$ and $v_l = \sqrt{\left[\exp\left(\sigma_l^2\right) - \delta_{l(1)}\right]/\delta_{l(1)}}$, for $i, l = 1, 2, \ldots, k$ and $i \neq l$, be the CVs of $X_i$ and $X_l$, respectively. In addition, let $\hat{v}_i$ and $\hat{v}_l$ be the estimators of $v_i$ and $v_l$, respectively. Let $L_{il} = \hat{v}_i - \hat{v}_l - \sqrt{(\hat{v}_i - l_i)^2 + (u_l - \hat{v}_l)^2}$ and $U_{il} = \hat{v}_i - \hat{v}_l + \sqrt{(u_i - \hat{v}_i)^2 + (\hat{v}_l - l_l)^2}$, where $i, l = 1, 2, \ldots, k$ and $i \neq l$, be the respective lower and upper limits of the confidence interval for $v_{il} = v_i - v_l$. Therefore,

$$P(L_{il} \leq v_{il} \leq U_{il}, \forall i \neq l) \to 1 - \alpha. \tag{27}$$

**Proof** Suppose that the respective lower and upper limits of the confidence interval for $v_{il} = v_i - v_l$ are

$$L_{il} = \hat{v}_i - \hat{v}_l - \sqrt{(\hat{v}_i - l_i)^2 + (u_l - \hat{v}_l)^2} = \hat{v}_{il} - \sqrt{(\hat{v}_i - l_i)^2 + (u_l - \hat{v}_l)^2}$$

and

$$U_{il} = \hat{v}_i - \hat{v}_l + \sqrt{(u_i - \hat{v}_i)^2 + (\hat{v}_l - l_l)^2} = \hat{v}_{il} + \sqrt{(u_i - \hat{v}_i)^2 + (\hat{v}_l - l_l)^2},$$

where $i, l = 1, 2, \ldots, k$ and $i \neq l$. Thus, the respective estimators of variance for $\hat{v}_i$ and $\hat{v}_l$ at $v_i = l_i$ and $v_l = l_l$ are

$$\hat{Var}(\hat{v}_i) = \frac{(\hat{v}_i - l_i)^2}{z_{\alpha/2}^2}$$

and

$$\hat{Var}(\hat{v}_l) = \frac{(\hat{v}_l - l_l)^2}{z_{\alpha/2}^2},$$

where $z_{\alpha/2}$ is the $\alpha/2$-th quantile of the standard normal distribution. Similarly, the respective estimators of variance for $\hat{v}_i$ and $\hat{v}_l$ at $v_i = u_i$ and $v_l = u_l$ are

$$\hat{Var}(\hat{v}_i) = \frac{(u_i - \hat{v}_i)^2}{z_{\alpha/2}^2}$$

and

$$\hat{Var}(\hat{v}_l) = \frac{(u_l - \hat{v}_l)^2}{z_{\alpha/2}^2}.$$

Hence, the respective lower and upper limits can be expressed as

$$L_{il} = \hat{v}_{il} - z_{\alpha/2}\sqrt{\frac{(\hat{v}_i - l_i)^2}{z_{\alpha/2}^2} + \frac{(u_l - \hat{v}_l)^2}{z_{\alpha/2}^2}}$$

$$= \hat{v}_{il} - z_{\alpha/2}\sqrt{\hat{Var}(\hat{v}_i) + \hat{Var}(\hat{v}_l)}$$

and

$$U_{il} = \hat{v}_{il} + z_{\alpha/2}\sqrt{\frac{(u_i - \hat{v}_i)^2}{z_{\alpha/2}^2} + \frac{(\hat{v}_l - l_l)^2}{z_{\alpha/2}^2}}$$

$$= \hat{v}_{il} + z_{\alpha/2}\sqrt{\hat{Var}(\hat{v}_i) + \hat{Var}(\hat{v}_l)}.$$

Therefore,

$$P(L_{il} \leq v_{il} \leq U_{il}) = P\left[v_{il} \in \left(\hat{v}_{il} \pm z_{\alpha/2}\sqrt{\hat{Var}(\hat{v}_i) + \hat{Var}(\hat{v}_l)}\right), \forall i \neq l\right]$$

$$= P\left[\max_{i \neq l}\left|\frac{\hat{v}_{il} - v_{il}}{\sqrt{\hat{Var}(\hat{v}_i) + \hat{Var}(\hat{v}_l)}}\right| \leq z_{\alpha/2}\right]$$

$$= P\left[D'_n \leq z_{\alpha/2}\right].$$

Suppose that $n_i/n \to r_i \in (0,1)$ as $n \to \infty, i = 1, 2, \ldots, k$ where $n = n_1 + n_2 + \ldots + n_k$. From the central limit theorem, $n(\hat{v}_i - v_i) \xrightarrow{d} Z_i, i = 1, 2, \ldots, k$, where $Z_i \overset{iid}{\sim} N\left(0, \sigma_i^2/r_i\right)$, while from Slutsky's theorem, $D'_n \to D'$, where $D' = \max_{i \neq l}\left|(Z_i - Z_l)/\sqrt{\sigma_i^2/r_i + \sigma_l^2/r_l}\right|$.

Following Skorokhod's theorem, let $Y_n$ and $Y$ be random variables from the common probability space with distributions $D'_n$ and $D'$, respectively. Thus, $Y_n$ converges to $Y$ almost surely, denoted by $Y_n \xrightarrow{a.s.} Y$, and $D'_n$ converges to $D'$ almost surely, denoted by $D'_n \xrightarrow{a.s.} D'$. Assume that $Z_i$ and $Z_i^*$ are independent and identically distributed random variables. Thus, $T(X, X^*, \mu, \sigma^2) \to D'^*$, where $D'^* = \max_{i \neq l}\left|\left(Z_i^* - Z_l^*/\sqrt{\sigma_i^2/r_i + \sigma_l^2/r_l}\right)\right|$, for $i, l = 1, 2, \ldots, k$, and $i \neq l$. Since the limiting distribution of $T(X, X^*, \mu, \sigma^2)$ is continuous and $z_{\alpha/2}(X) \to q_{\alpha/2}$, where $q_{\alpha/2}$ is the $\alpha/2$-th quantile of the distribution of $D'^*$, we can obtain

$$P\left(D'_n \leq z_{\alpha/2}\right) \to P\left(D' \leq q_{\alpha/2}\right) = P\left(D'^* \leq q_{\alpha/2}\right) = 1 - \alpha, \text{ as } n \to \infty. \text{ Therefore,}$$

$$P\left[v_{il} \in \left(\hat{v}_{il} \pm z_{\alpha/2}\sqrt{\hat{Var}(\hat{v}_i) + \hat{Var}(\hat{v}_l)}, \forall i \neq l\right)\right] \to 1 - \alpha,$$

which implies that

$$P(L_{il} \leq v_{il} \leq U_{il}, \forall i \neq l) \to 1 - \alpha. \quad \square$$

**Algorithm 2:** For MOVER
Step 1.  Generate random samples $X_i, i = 1, 2, \ldots, k$ with sample size $n_1, n_2, \ldots, n_k$ and calculate $\hat{\delta}_{i(1)}$ and $\hat{\sigma}_i^2$.
Step 2.  Generate $\chi^2_{1-\alpha/2, n_{i(1)}-1}, \chi^2_{\alpha/2, n_{i(1)}-1}$, and $Z_i \sim N(0,1)$.
Step 3.  Calculate $l_{\sigma_i^2}, l_{\sigma_l^2}, u_{\sigma_i^2}, u_{\sigma_l^2}, l_{\delta_{i(1)}}, l_{\delta_{l(1)}}, u_{\delta_{i(1)}}$, and $u_{\delta_{l(1)}}$.
Step 4.  Repeat Steps 2–3 5,000 times.
Step 5.  Compute the 95% SCIs for $v_{il}$.
Step 6.  Repeat Steps 1–5 15,000 times.

## SIMULATION RESULTS

Here, the performances of the proposed methods via Monte Carlo simulation with the R statistical program are presented. The best method attains a coverage probability equal to or greater than the nominal simultaneous confidence level of 0.95 together with the

shortest expected length. The simulations were conducted with 15,000 iterations for each combination of parameters. Furthermore, 5,000 replications for the FGCI and Bayesian methods for each case of parameter combination were carried out. Sample sizes were set as 25, 50, and 100; $\delta_{i(1)} = 0.2, 0.5, 0.8$; and $\sigma_i^2 = 0.5, 1.0, 2.0$.

The results in Tables 2–4 and Figs. 1–3 show that the coverage probabilities of FGCI and the equal-tailed Bayesian using Jeffreys' rule prior were close to or greater than the nominal confidence level for almost all $k$ values. Similarly, the coverage probabilities of the equal-tailed Bayesian using the uniform prior, the Bayesian credible intervals using Jeffreys' rule and uniform priors, and MOVER were close to or greater than the nominal confidence level for all cases. For most cases, the Bayesian credible interval using Jeffreys' rule prior attained the shortest expected length, except for $n_i = 50$; $\delta_{i(1)} = 0.5, 0.8$; and $\sigma_i^2 = 0.5, 1.0$, for which the expected lengths of FGCI were the shortest.

## EMPIRICAL STUDY

Thailand is generally divided into five areas by topography, i.e., Northern (A1), Northeastern (A2), Central (A3), Eastern (A4), and Southern (A5). The daily rainfall data from these areas in August 2020 were used to assess the performances of the proposed methods for SCI construction. The distributions of these data are presented in Fig. 4, which shows right-skewness for all of the datasets. Thus, the minimum Akaike information criterion (AIC) and the lowest Bayesian information criterion (BIC) were used to test the fitting of the distributions to such data. From AIC and BIC results in Table 5, it is evident that the positive values in the rainfall datasets from the five areas conform to lognormal distributions. Moreover, normal Q-Q plots were constructed to show the distributions of the log-transformed positive rainfall data from the five areas (Fig. 5), which verified the AIC and BIC results that these datasets follow lognormal distributions. A summary of these data are

$$n_1 = 31, \hat{\delta}_1 = 0.7097, \hat{\mu}_1 = 0.7715, \hat{\sigma}_1^2 = 3.4565, \hat{\eta}_1 = 6.6088,$$
$$n_2 = 31, \hat{\delta}_2 = 0.6774, \hat{\mu}_2 = 1.4332, \hat{\sigma}_2^2 = 2.9550, \hat{\eta}_2 = 5.2294,$$
$$n_3 = 31, \hat{\delta}_3 = 0.6452, \hat{\mu}_3 = 1.5512, \hat{\sigma}_3^2 = 2.8638, \hat{\eta}_3 = 5.1154,$$
$$n_4 = 31, \hat{\delta}_4 = 0.4839, \hat{\mu}_4 = 1.4178, \hat{\sigma}_4^2 = 2.1487, \hat{\eta}_4 = 4.0888,$$
$$n_5 = 31, \hat{\delta}_5 = 0.4839, \hat{\mu}_5 = 1.8040, \hat{\sigma}_5^2 = 2.1962, \hat{\eta}_5 = 4.1930.$$

Table 6 reports the 95% SCIs and credible intervals for all pairwise differences between the CVs of the daily rainfall series from five areas in Thailand. The results show that the expected length of the Bayesian credible interval using the Jeffreys' rule prior was the shortest, which corresponds with the simulation results. Therefore, it is a good choice for constructing the SCI for all of the pairwise differences between the CVs of the precipitation series from the five areas in Thailand.

## DISCUSSION

The simulation results indicate that the Bayesian credible interval using Jeffreys' rule prior outperformed the other methods in virtually all cases. Although the coverage probabilities

**Table 2** The coverage probabilities and expected lengths for the 95% SCIs and credible intervals for all pairwise differences between the CVs of delta-lognormal distributions for $k = 3$.

| $n_1 : n_2 : n_3$ | $\delta_{1(1)} : \delta_{2(1)} : \delta_{3(1)}$ | $\sigma_1^2 : \sigma_2^2 : \sigma_3^2$ | Coverage probabilities (Expected lengths) | | | | | |
|---|---|---|---|---|---|---|---|---|
| | | | FGCI | B.Jrule-E | B.Uni-E | B.Jrule-C | B.Uni-C | MOVER |
| 25:25:25 | 0.5:0.5:0.5 | 0.5:0.5:0.5 | 0.9642 | 0.9788 | 0.9842 | 0.9956 | 0.9980 | 0.9986 |
| | | | (2.1698) | (2.1537) | (2.5030) | (2.0848) | (2.4026) | (3.4408) |
| | | 1.0:1.0:1.0 | 0.9573 | 0.9605 | 0.9718 | 0.9957 | 0.9984 | 0.9941 |
| | | | (6.8818) | (6.2442) | (7.9188) | (5.7177) | (7.1021) | (9.6034) |
| | | 2.0:2.0:2.0 | 0.9516 | 0.9465 | 0.9631 | 0.9978 | 0.9991 | 0.9838 |
| | | | (68.0142) | (53.9752) | (88.1169) | (37.9389) | (55.1280) | (85.1444) |
| | | 0.5:1.0:2.0 | 0.9557 | 0.9540 | 0.9678 | 0.9708 | 0.9820 | 0.9878 |
| | | | (24.7584) | (20.5209) | (31.5389) | (13.1644) | (17.7557) | (31.4612) |
| | 0.8:0.8:0.8 | 0.5:0.5:0.5 | 0.9542 | 0.9632 | 0.9725 | 0.9834 | 0.9894 | 0.9954 |
| | | | (1.2558) | (1.2613) | (1.3684) | (1.2412) | (1.3444) | (1.8437) |
| | | 1.0:1.0:1.0 | 0.9533 | 0.9526 | 0.9636 | 0.9857 | 0.9915 | 0.9868 |
| | | | (3.2363) | (3.1117) | (3.4638) | (2.9940) | (3.3202) | (4.2113) |
| | | 2.0:2.0:2.0 | 0.9499 | 0.9458 | 0.9570 | 0.9930 | 0.9963 | 0.9756 |
| | | | (16.6440) | (15.4579) | (18.2078) | (13.7933) | (16.0213) | (20.0584) |
| | | 0.5:1.0:2.0 | 0.9513 | 0.9496 | 0.9598 | 0.9659 | 0.9750 | 0.9797 |
| | | | (7.3206) | (6.8777) | (7.9629) | (5.5921) | (6.3072) | (8.7800) |
| 50:50:50 | 0.2:0.2:0.2 | 0.5:0.5:0.5 | 0.9692 | 0.9869 | 0.9906 | 0.9991 | 0.9997 | 0.9994 |
| | | | (4.3383) | (4.1487) | (5.2706) | (3.9321) | (4.8731) | (7.2083) |
| | | 1.0:1.0:1.0 | 0.9593 | 0.9672 | 0.9778 | 0.9988 | 0.9996 | 0.9965 |
| | | | (17.7495) | (14.6338) | (23.0519) | (12.3745) | (17.8689) | (25.8720) |
| | | 2.0:2.0:2.0 | 0.9525 | 0.9489 | 0.9668 | 0.9984 | 0.9996 | 0.9882 |
| | | | (813.5319) | (360.7209) | (3.75E+03) | (130.6376) | (338.9741) | (1.00E+03) |
| | | 0.5:1.0:2.0 | 0.9560 | 0.9567 | 0.9719 | 0.9742 | 0.9846 | 0.9910 |
| | | | (131.2015) | (80.4363) | (238.4643) | (36.1680) | (67.2572) | (169.2385) |
| | 0.5:0.5:0.5 | 0.5:0.5:0.5 | 0.9609 | 0.9797 | 0.9827 | 0.9904 | 0.9926 | 0.9990 |
| | | | (1.2086) | (1.2989) | (1.3657) | (1.2849) | (1.3499) | (1.9562) |
| | | 1.0:1.0:1.0 | 0.9536 | 0.9613 | 0.9678 | 0.9870 | 0.9907 | 0.9934 |
| | | | (3.0015) | (2.9770) | (3.1957) | (2.8997) | (3.1064) | (4.2360) |
| | | 2.0:2.0:2.0 | 0.9496 | 0.9488 | 0.9579 | 0.9910 | 0.9942 | 0.9831 |
| | | | (13.0784) | (12.5308) | (13.8315) | (11.6629) | (12.8048) | (16.6567) |
| | | 0.5:1.0:2.0 | 0.9510 | 0.9541 | 0.9619 | 0.9644 | 0.9712 | 0.9872 |
| | | | (6.2117) | (6.0214) | (6.5869) | (5.1934) | (5.6079) | (7.9402) |
| | 0.8:0.8:0.8 | 0.5:0.5:0.5 | 0.9545 | 0.9652 | 0.9702 | 0.9764 | 0.9804 | 0.9960 |
| | | | (0.7563) | (0.7866) | (0.8128) | (0.7810) | (0.8069) | (1.1146) |
| | | 1.0:1.0:1.0 | 0.9512 | 0.9530 | 0.9582 | 0.9744 | 0.9788 | 0.9870 |
| | | | (1.7222) | (1.7159) | (1.7838) | (1.6925) | (1.7587) | (2.2244) |
| | | 2.0:2.0:2.0 | 0.9489 | 0.9476 | 0.9528 | 0.9832 | 0.9863 | 0.9746 |
| | | | (6.2458) | (6.1250) | (6.4310) | (5.9199) | (6.2078) | (7.4204) |
| | | 0.5:1.0:2.0 | 0.9534 | 0.9532 | 0.9592 | 0.9644 | 0.9690 | 0.9816 |

**Table 2** (*continued*)

| $n_1 : n_2 : n_3$ | $\delta_{1(1)} : \delta_{2(1)} : \delta_{3(1)}$ | $\sigma_1^2 : \sigma_2^2 : \sigma_3^2$ | Coverage probabilities (Expected lengths) | | | | | |
|---|---|---|---|---|---|---|---|---|
| | | | FGCI | B.Jrule-E | B.Uni-E | B.Jrule-C | B.Uni-C | MOVER |
| | | | (3.2059) | (3.1646) | (3.3108) | (2.8957) | (3.0156) | (3.8254) |
| 100:100:100 | 0.2:0.2:0.2 | 0.5:0.5:0.5 | 0.9655 | 0.9862 | 0.9876 | 0.9955 | 0.9969 | 0.9995 |
| | | | (2.1032) | (2.2970) | (2.4455) | (2.2649) | (2.4065) | (3.5694) |
| | | 1.0:1.0:1.0 | 0.9572 | 0.9696 | 0.9746 | 0.9935 | 0.9959 | 0.9966 |
| | | | (5.5368) | (5.4606) | (6.0091) | (5.2547) | (5.7570) | (8.2473) |
| | | 2.0:2.0:2.0 | 0.9528 | 0.9546 | 0.9629 | 0.9951 | 0.9973 | 0.9891 |
| | | | (28.1449) | (26.2788) | (30.4653) | (23.6063) | (27.0268) | (37.4768) |
| | | 0.5:1.0:2.0 | 0.9541 | 0.9585 | 0.9665 | 0.9672 | 0.9744 | 0.9903 |
| | | | (12.6988) | (12.0766) | (13.7614) | (9.9027) | (11.0231) | (16.9562) |
| | 0.5:0.5:0.5 | 0.5:0.5:0.5 | 0.9595 | 0.9793 | 0.9803 | 0.9852 | 0.9866 | 0.9988 |
| | | | (0.7728) | (0.8596) | (0.8774) | (0.8547) | (0.8723) | (1.2568) |
| | | 1.0:1.0:1.0 | 0.9558 | 0.9646 | 0.9673 | 0.9806 | 0.9824 | 0.9949 |
| | | | (1.7356) | (1.7761) | (1.8252) | (1.7576) | (1.8058) | (2.4523) |
| | | 2.0:2.0:2.0 | 0.9501 | 0.9513 | 0.9555 | 0.9803 | 0.9831 | 0.9832 |
| | | | (6.1101) | (6.0586) | (6.2714) | (5.9140) | (6.1174) | (7.6906) |
| | | 0.5:1.0:2.0 | 0.9518 | 0.9565 | 0.9603 | 0.9613 | 0.9653 | 0.9889 |
| | | | (3.1854) | (3.1976) | (3.2977) | (2.9846) | (3.0716) | (4.0742) |
| | 0.8:0.8:0.8 | 0.5:0.5:0.5 | 0.9542 | 0.9668 | 0.9691 | 0.9721 | 0.9739 | 0.9966 |
| | | | (0.5013) | (0.5298) | (0.5376) | (0.5273) | (0.5350) | (0.7395) |
| | | 1.0:1.0:1.0 | 0.9495 | 0.9526 | 0.9554 | 0.9654 | 0.9678 | 0.9864 |
| | | | (1.0816) | (1.0919) | (1.1103) | (1.0843) | (1.1025) | (1.3892) |
| | | 2.0:2.0:2.0 | 0.9517 | 0.9516 | 0.9546 | 0.9739 | 0.9756 | 0.9766 |
| | | | (3.4703) | (3.4545) | (3.5212) | (3.4088) | (3.4741) | (4.0656) |
| | | 0.5:1.0:2.0 | 0.9496 | 0.9518 | 0.9552 | 0.9579 | 0.9608 | 0.9814 |
| | | | (1.8867) | (1.8883) | (1.9231) | (1.8062) | (1.8377) | (2.2430) |
| 25:50:100 | 0.5:0.5:0.5 | 0.5:0.5:0.5 | 0.9614 | 0.9782 | 0.9840 | 0.9829 | 0.9885 | 0.9984 |
| | | | (1.4118) | (1.4647) | (1.6172) | (1.4069) | (1.5307) | (2.2374) |
| | | 1.0:1.0:1.0 | 0.9525 | 0.9572 | 0.9670 | 0.9764 | 0.9838 | 0.9941 |
| | | | (3.9253) | (3.7120) | (4.3712) | (3.3455) | (3.7992) | (5.4323) |
| | | 2.0:2.0:2.0 | 0.9525 | 0.9502 | 0.9609 | 0.9827 | 0.9875 | 0.9826 |
| | | | (27.5125) | (23.2226) | (33.9099) | (16.5621) | (20.9132) | (34.6995) |
| | | 0.5:1.0:2.0 | 0.9551 | 0.9600 | 0.9648 | 0.9793 | 0.9827 | 0.9901 |
| | | | (3.8830) | (3.8512) | (4.0951) | (3.6913) | (3.9292) | (5.2078) |
| 25:50:100 | 0.8:0.8:0.8 | 0.5:0.5:0.5 | 0.9572 | 0.9679 | 0.9740 | 0.9754 | 0.9825 | 0.9958 |
| | | | (0.8600) | (0.8810) | (0.9323) | (0.8589) | (0.9050) | (1.2513) |
| | | 1.0:1.0:1.0 | 0.9516 | 0.9525 | 0.9596 | 0.9709 | 0.9775 | 0.9869 |
| | | | (2.0776) | (2.0320) | (2.1907) | (1.9232) | (2.0528) | (2.6582) |
| | | 2.0:2.0:2.0 | 0.9513 | 0.9486 | 0.9558 | 0.9776 | 0.9826 | 0.9751 |
| | | | (8.8667) | (8.4294) | (9.4811) | (7.3245) | (8.0262) | (10.4756) |
| | | 0.5:1.0:2.0 | 0.9532 | 0.9545 | 0.9592 | 0.9702 | 0.9735 | 0.9833 |
| | | | (2.2424) | (2.2348) | (2.3100) | (2.1757) | (2.2505) | (2.7807) |

Notes.
B.Jrule-E, B.Uni-E represented the equal-tailed Bayesian confidence intervals using Jeffreys' rule and uniform priors, respectively, and B.Jrule-C and B.Uni-C represented the Bayesian credible intervals using Jeffrey's rule and uniform priors.

**Table 3** The coverage probabilities and expected lengths for the 95% SCIs and credible intervals for all pairwise differences between the CVs of delta-lognormal distributions for $k = 5$.

| $n_1 : \ldots : n_5$ | $\delta_{1(1)} : \ldots : \delta_{5(1)}$ | $\sigma_1^2 : \ldots : \sigma_5^2$ | Coverage probabilities (Expected lengths) | | | | | |
|---|---|---|---|---|---|---|---|---|
| | | | FGCI | B.Jrule-E | B.Uni-E | B.Jrule-C | B.Uni-C | MOVER |
| $25^5$ | $0.5^5$ | $0.5^5$ | 0.9643 | 0.9794 | 0.9854 | 0.9958 | 0.9981 | 0.9986 |
| | | | (2.1555) | (2.1428) | (2.4893) | (2.0756) | (2.3915) | (3.4245) |
| | | $1.0^5$ | 0.9558 | 0.9597 | 0.9717 | 0.9954 | 0.9982 | 0.9935 |
| | | | (6.9019) | (6.2584) | (7.9374) | (5.7313) | (7.1150) | (9.6234) |
| | | $2.0^5$ | 0.9519 | 0.9470 | 0.9625 | 0.9973 | 0.9993 | 0.9834 |
| | | | (62.1565) | (50.5751) | (80.8805) | (37.1890) | (53.9109) | (79.5531) |
| | | $0.5^2 : 1.0 : 2.0^2$ | 0.9545 | 0.9538 | 0.9676 | 0.9715 | 0.9818 | 0.9870 |
| | | | (25.5380) | (23.5178) | (37.1670) | (15.2344) | (20.9148) | (36.4956) |
| | $0.8^5$ | $0.5^5$ | 0.9548 | 0.9642 | 0.9728 | 0.9833 | 0.9895 | 0.9958 |
| | | | (1.2560) | (1.2612) | (1.3683) | (1.2413) | (1.3445) | (1.8432) |
| | | $1.0^5$ | 0.9529 | 0.9523 | 0.9627 | 0.9857 | 0.9915 | 0.9875 |
| | | | (3.2430) | (3.1186) | (3.4726) | (3.0017) | (3.3297) | (4.2211) |
| | | $2.0^5$ | 0.9495 | 0.9447 | 0.9567 | 0.9929 | 0.9963 | 0.9757 |
| | | | (16.3407) | (15.2264) | (17.9101) | (13.5908) | (15.7761) | (19.7367) |
| | | $0.5^2 : 1.0 : 2.0^2$ | 0.9525 | 0.9508 | 0.9614 | 0.9696 | 0.9775 | 0.9807 |
| | | | (7.2308) | (7.5393) | (8.7262) | (6.1994) | (7.0193) | (9.6191) |
| $50^5$ | $0.2^5$ | $0.5^5$ | 0.9676 | 0.9864 | 0.9896 | 0.9990 | 0.9996 | 0.9994 |
| | | | (4.3044) | (4.1252) | (5.2361) | (3.9094) | (4.8405) | (7.1615) |
| | | $1.0^5$ | 0.9610 | 0.9688 | 0.9792 | 0.9988 | 0.9997 | 0.9969 |
| | | | (17.6359) | (14.5147) | (22.9054) | (12.3017) | (17.7946) | (25.7023) |
| | | $2.0^5$ | 0.9532 | 0.9499 | 0.9673 | 0.9988 | 0.9997 | 0.9882 |
| | | | (428.000) | (253.198) | (996.537) | (122.265) | (272.000) | (578.397) |
| | | $0.5^2 : 1.0 : 2.0^2$ | 0.9548 | 0.9550 | 0.9707 | 0.9735 | 0.9849 | 0.9903 |
| | | | (191.069) | (122.152) | (490.477) | (47.626) | (101.477) | (276.993) |
| | $0.5^5$ | $0.5^5$ | 0.9602 | 0.9789 | 0.9818 | 0.9899 | 0.9921 | 0.9987 |
| | | | (1.2089) | (1.2989) | (1.3660) | (1.2850) | (1.3502) | (1.9561) |
| | | $1.0^5$ | 0.9548 | 0.9618 | 0.9677 | 0.9876 | 0.9911 | 0.9941 |
| | | | (3.0083) | (2.9852) | (3.2034) | (2.9072) | (3.1133) | (4.2457) |
| | | $2.0^5$ | 0.9506 | 0.9496 | 0.9578 | 0.9913 | 0.9944 | 0.9839 |
| | | | (13.0787) | (12.5396) | (13.8536) | (11.6771) | (12.8287) | (16.6737) |
| | | $0.5^2 : 1.0 : 2.0^2$ | 0.9537 | 0.9574 | 0.9644 | 0.9695 | 0.9752 | 0.9881 |
| | | | (6.1486) | (6.5705) | (7.1952) | (5.7079) | (6.1752) | (8.6478) |
| | $0.8^5$ | $0.5^5$ | 0.9541 | 0.9654 | 0.9700 | 0.9763 | 0.9799 | 0.9958 |
| | | | (0.7579) | (0.7881) | (0.8142) | (0.7825) | (0.8083) | (1.1163) |
| | | $1.0^5$ | 0.9516 | 0.9536 | 0.9589 | 0.9755 | 0.9790 | 0.9874 |
| | | | (1.7233) | (1.7170) | (1.7855) | (1.6937) | (1.7603) | (2.2261) |
| | | $2.0^5$ | 0.9508 | 0.9495 | 0.9548 | 0.9838 | 0.9871 | 0.9760 |
| | | | (6.2622) | (6.1471) | (6.4545) | (5.9443) | (6.2338) | (7.4475) |
| | | $0.5^2 : 1.0 : 2.0^2$ | 0.9512 | 0.9522 | 0.9574 | 0.9646 | 0.9691 | 0.9808 |

Table 3 (*continued*)

| $n_1 : \ldots : n_5$ | $\delta_{1(1)} : \ldots : \delta_{5(1)}$ | $\sigma_1^2 : \ldots : \sigma_5^2$ | Coverage probabilities (Expected lengths) | | | | | |
| --- | --- | --- | --- | --- | --- | --- | --- | --- |
| | | | FGCI | B.Jrule-E | B.Uni-E | B.Jrule-C | B.Uni-C | MOVER |
| | | | (3.1366) | (3.3900) | (3.5463) | (3.1199) | (3.2508) | (4.0931) |
| $100^5$ | $0.2^5$ | $0.5^5$ | 0.9660 | 0.9867 | 0.9883 | 0.9954 | 0.9966 | 0.9994 |
| | | | (2.1027) | (2.2959) | (2.4445) | (2.2638) | (2.4055) | (3.5676) |
| | | $1.0^5$ | 0.9566 | 0.9680 | 0.9732 | 0.9935 | 0.9958 | 0.9968 |
| | | | (5.5583) | (5.4771) | (6.0303) | (5.2701) | (5.7768) | (8.2755) |
| | | $2.0^5$ | 0.9521 | 0.9526 | 0.9617 | 0.9951 | 0.9971 | 0.9883 |
| | | | (27.9967) | (26.1432) | (30.2853) | (23.4497) | (26.8163) | (37.2629) |
| | | $0.5^2 : 1.0 : 2.0^2$ | 0.9550 | 0.9598 | 0.9676 | 0.9697 | 0.9763 | 0.9915 |
| | | | (12.6571) | (13.2932) | (15.2030) | (11.0107) | (12.3119) | (18.6781) |
| | $0.5^5$ | $0.5^5$ | 0.9594 | 0.9791 | 0.9807 | 0.9848 | 0.9862 | 0.9987 |
| | | | (0.7720) | (0.8589) | (0.8767) | (0.8540) | (0.8717) | (1.2557) |
| | | $1.0^5$ | 0.9546 | 0.9635 | 0.9664 | 0.9793 | 0.9816 | 0.9947 |
| | | | (1.7331) | (1.7739) | (1.8229) | (1.7555) | (1.8035) | (2.4495) |
| | | $2.0^5$ | 0.9527 | 0.9540 | 0.9582 | 0.9825 | 0.9852 | 0.9848 |
| | | | (6.0963) | (6.0429) | (6.2535) | (5.9008) | (6.1028) | (7.6733) |
| | | $0.5^2 : 1.0 : 2.0^2$ | 0.9527 | 0.9579 | 0.9613 | 0.9645 | 0.9681 | 0.9884 |
| | | | (3.1078) | (3.4067) | (3.5167) | (3.1961) | (3.2916) | (4.3313) |
| $100^5$ | $0.8^5$ | $0.5^5$ | 0.9542 | 0.9670 | 0.9691 | 0.9721 | 0.9742 | 0.9966 |
| | | | (0.5010) | (0.5295) | (0.5374) | (0.5269) | (0.5348) | (0.7392) |
| | | $1.0^5$ | 0.9509 | 0.9544 | 0.9570 | 0.9665 | 0.9688 | 0.9875 |
| | | | (1.0812) | (1.0918) | (1.1102) | (1.0842) | (1.1024) | (1.3890) |
| | | $2.0^5$ | 0.9506 | 0.9507 | 0.9533 | 0.9730 | 0.9752 | 0.9755 |
| | | | (3.4721) | (3.4569) | (3.5238) | (3.4111) | (3.4762) | (4.0692) |
| | | $0.5^2 : 1.0 : 2.0^2$ | 0.9500 | 0.9524 | 0.9552 | 0.9593 | 0.9619 | 0.9806 |
| | | | (1.8380) | (2.0041) | (2.0416) | (1.9233) | (1.9575) | (2.3760) |
| $25^2 : 50 : 100^2$ | $0.5^5$ | $0.5^5$ | 0.9608 | 0.9770 | 0.9830 | 0.9830 | 0.9886 | 0.9985 |
| | | | (1.5163) | (1.4923) | (1.6614) | (1.4314) | (1.5704) | (2.2909) |
| | | $1.0^5$ | 0.9549 | 0.9595 | 0.9688 | 0.9785 | 0.9852 | 0.9940 |
| | | | (4.4104) | (3.8914) | (4.6543) | (3.4894) | (4.0239) | (5.7364) |
| | | $2.0^5$ | 0.9511 | 0.9484 | 0.9600 | 0.9813 | 0.9872 | 0.9826 |
| | | | (32.7160) | (25.3223) | (37.4111) | (17.8405) | (23.1773) | (37.9900) |
| | | $0.5^2 : 1.0 : 2.0^2$ | 0.9552 | 0.9603 | 0.9655 | 0.9801 | 0.9835 | 0.9901 |
| | | | (3.8324) | (4.0083) | (4.2595) | (3.8460) | (4.0906) | (5.3939) |
| | $0.8^5$ | $0.5^5$ | 0.9555 | 0.9656 | 0.9719 | 0.9753 | 0.9817 | 0.9959 |
| | | | (0.9159) | (0.8965) | (0.9522) | (0.8736) | (0.9239) | (1.2760) |
| | | $1.0^5$ | 0.9506 | 0.9512 | 0.9583 | 0.9707 | 0.9774 | 0.9856 |
| | | | (2.2438) | (2.0817) | (2.2545) | (1.9682) | (2.1113) | (2.7304) |
| | | $2.0^5$ | 0.9496 | 0.9469 | 0.9551 | 0.9774 | 0.9824 | 0.9744 |
| | | | (10.1646) | (8.9783) | (10.2014) | (7.7693) | (8.6055) | (11.2231) |
| | | $0.5^2 : 1.0 : 2.0^2$ | 0.9515 | 0.9536 | 0.9582 | 0.9704 | 0.9737 | 0.9832 |
| | | | (2.2063) | (2.3189) | (2.3964) | (2.2595) | (2.3367) | (2.8768) |

Notes.
$25^5$ represents 25:25:25:25:25.
**Table 4 The coverage probabilities and expected lengths for the 95% SCIs and credible intervals for all pairwise differences between the CVs of delta-lognormal distributions for $k = 10$.**

| $n_1:\ldots:n_{10}$ | $\delta_{1(1)}:\ldots:\delta_{10(1)}$ | $\sigma_1^2:\ldots:\sigma_{10}^2$ | Coverage probabilities (Expected lengths) | | | | | |
|---|---|---|---|---|---|---|---|---|
| | | | FGCI | B.Jrule-E | B.Uni-E | B.Jrule-C | B.Uni-C | MOVER |
| $25^{10}$ | $0.5^{10}$ | $0.5^{10}$ | 0.9644 | 0.9796 | 0.9852 | 0.9958 | 0.9981 | 0.9987 |
| | | | (2.1718) | (2.1549) | (2.5017) | (2.0864) | (2.4056) | (3.4437) |
| | | $1.0^{10}$ | 0.9559 | 0.9596 | 0.9717 | 0.9952 | 0.9982 | 0.9935 |
| | | | (6.9095) | (6.2680) | (7.9343) | (5.7342) | (7.1201) | (9.6381) |
| | | $2.0^{10}$ | 0.9513 | 0.9466 | 0.9627 | 0.9970 | 0.9990 | 0.9832 |
| | | | (64.2109) | (51.6243) | (83.5403) | (37.5819) | (54.7163) | (81.8119) |
| | $0.8^{10}$ | $0.5^{10}$ | 0.9558 | 0.9645 | 0.9733 | 0.9836 | 0.9899 | 0.9958 |
| | | | (1.2509) | (1.2566) | (1.3621) | (1.2369) | (1.3398) | (1.8373) |
| | | $1.0^{10}$ | 0.9518 | 0.9513 | 0.9621 | 0.9855 | 0.9914 | 0.9871 |
| | | | (3.2511) | (3.1246) | (3.4760) | (3.0068) | (3.3358) | (4.2290) |
| | | $2.0^{10}$ | 0.9511 | 0.9468 | 0.9581 | 0.9931 | 0.9962 | 0.9759 |
| | | | (16.2516) | (15.1258) | (17.7445) | (13.5223) | (15.6914) | (19.6085) |
| $50^{10}$ | $0.2^{10}$ | $0.5^{10}$ | 0.9678 | 0.9859 | 0.9892 | 0.9988 | 0.9995 | 0.9993 |
| | | | (4.3466) | (4.1581) | (5.2771) | (3.9368) | (4.8806) | (7.2207) |
| | | $1.0^{10}$ | 0.9600 | 0.9680 | 0.9784 | 0.9988 | 0.9997 | 0.9966 |
| | | | (17.6759) | (14.5839) | (22.8034) | (12.3672) | (17.8805) | (25.7964) |
| | | $2.0^{10}$ | 0.9525 | 0.9491 | 0.9671 | 0.9988 | 0.9997 | 0.9881 |
| | | | (584.809) | (314.782) | (1958.586) | (130.579) | (314.105) | (825.014) |
| | $0.5^{10}$ | $0.5^{10}$ | 0.9614 | 0.9800 | 0.9828 | 0.9905 | 0.9927 | 0.9989 |
| | | | (1.2078) | (1.2981) | (1.3643) | (1.2842) | (1.3493) | (1.9553) |
| | | $1.0^{10}$ | 0.9543 | 0.9614 | 0.9676 | 0.9874 | 0.9909 | 0.9942 |
| | | | (2.9985) | (2.9750) | (3.1898) | (2.8975) | (3.1028) | (4.2323) |
| | | $2.0^{10}$ | 0.9505 | 0.9497 | 0.9577 | 0.9915 | 0.9944 | 0.9834 |
| | | | (13.0586) | (12.5061) | (13.7967) | (11.6493) | (12.7980) | (16.6309) |
| | $0.8^{10}$ | $0.5^{10}$ | 0.9543 | 0.9653 | 0.9699 | 0.9761 | 0.9799 | 0.9960 |
| | | | (0.7572) | (0.7873) | (0.8131) | (0.7817) | (0.8075) | (1.1153) |
| | | $1.0^{10}$ | 0.9510 | 0.9530 | 0.9584 | 0.9747 | 0.9791 | 0.9872 |
| | | | (1.7261) | (1.7192) | (1.7869) | (1.6958) | (1.7624) | (2.2287) |
| | | $2.0^{10}$ | 0.9500 | 0.9485 | 0.9541 | 0.9835 | 0.9866 | 0.9749 |
| | | | (6.2757) | (6.1553) | (6.4591) | (5.9506) | (6.2399) | (7.4567) |
| $100^{10}$ | $0.2^{10}$ | $0.5^{10}$ | 0.9658 | 0.9868 | 0.9885 | 0.9957 | 0.9968 | 0.9994 |
| | | | (2.1038) | (2.2979) | (2.4450) | (2.2660) | (2.4078) | (3.5707) |
| | | $1.0^{10}$ | 0.9573 | 0.9684 | 0.9738 | 0.9937 | 0.9959 | 0.9968 |
| | | | (5.5609) | (5.4793) | (6.0256) | (5.2730) | (5.7794) | (8.2780) |
| | | $2.0^{10}$ | 0.9515 | 0.9523 | 0.9612 | 0.9950 | 0.9972 | 0.9881 |
| | | | (27.8994) | (26.0553) | (30.1289) | (23.3884) | (26.7565) | (37.1355) |
| $100^{10}$ | $0.5^{10}$ | $0.5^{10}$ | 0.9595 | 0.9799 | 0.9813 | 0.9856 | 0.9869 | 0.9989 |
| | | | (0.7720) | (0.8589) | (0.8767) | (0.8540) | (0.8718) | (1.2559) |
| | | $1.0^{10}$ | 0.9536 | 0.9622 | 0.9653 | 0.9784 | 0.9810 | 0.9945 |
**Table 4** (*continued*)

| $n_1:\ldots:n_{10}$ | $\delta_{1(1)}:\ldots:\delta_{10(1)}$ | $\sigma_1^2:\ldots:\sigma_{10}^2$ | Coverage probabilities (Expected lengths) | | | | | |
|---|---|---|---|---|---|---|---|---|
| | | | **FGCI** | **B.Jrule-E** | **B.Uni-E** | **B.Jrule-C** | **B.Uni-C** | **MOVER** |
| | | | (1.7364) | (1.7768) | (1.8256) | (1.7582) | (1.8066) | (2.4531) |
| | | $2.0^{10}$ | 0.9505 | 0.9518 | 0.9557 | 0.9815 | 0.9841 | 0.9838 |
| | | | (6.1121) | (6.0620) | (6.2702) | (5.9173) | (6.1188) | (7.6931) |
| $0.8^{10}$ | $0.5^{10}$ | | 0.9541 | 0.9668 | 0.9689 | 0.9720 | 0.9741 | 0.9964 |
| | | | (0.5013) | (0.5298) | (0.5376) | (0.5272) | (0.5351) | (0.7395) |
| | $1.0^{10}$ | | 0.9508 | 0.9542 | 0.9569 | 0.9665 | 0.9689 | 0.9874 |
| | | | (1.0804) | (1.0907) | (1.1089) | (1.0832) | (1.1014) | (1.3878) |
| | $2.0^{10}$ | | 0.9503 | 0.9502 | 0.9530 | 0.9726 | 0.9747 | 0.9753 |
| | | | (3.4656) | (3.4492) | (3.5157) | (3.4036) | (3.4692) | (4.0610) |

Notes.

$25^{10}$ represents 25:25:25:25:25:25:25:25:25:25.

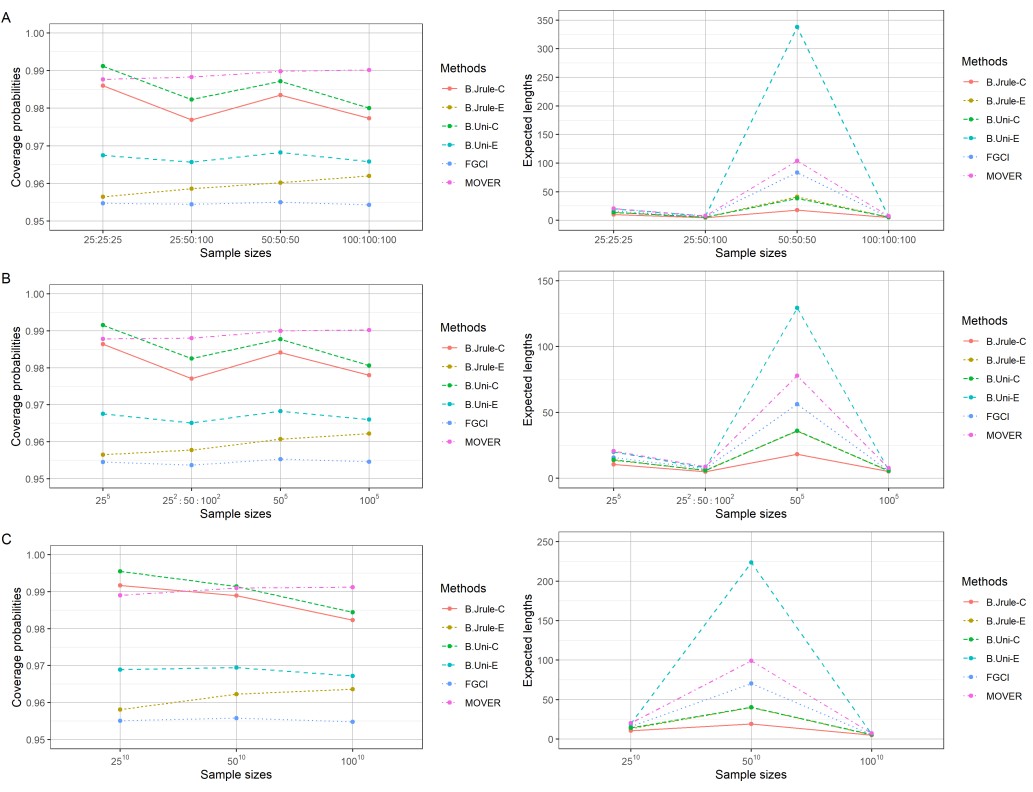

**Figure 1** Comparison of the performances of the proposed methods in terms of their coverage probabilities and expected lengths with various sample sizes: (A) $k = 3$ (B) $k = 5$ (C) $k = 10$.

in some cases were close to 1.00, suggesting that overestimation may have occurred, the expected lengths were the shortest. Therefore, the Bayesian credible interval using Jeffreys' rule prior can be used to construct the SCIs for all of the pairwise differences between the CVs of delta-lognormal distributions. Since constructing SCIs concerns the differences between the parameters of interest for all pairwise comparisons, our findings correspond

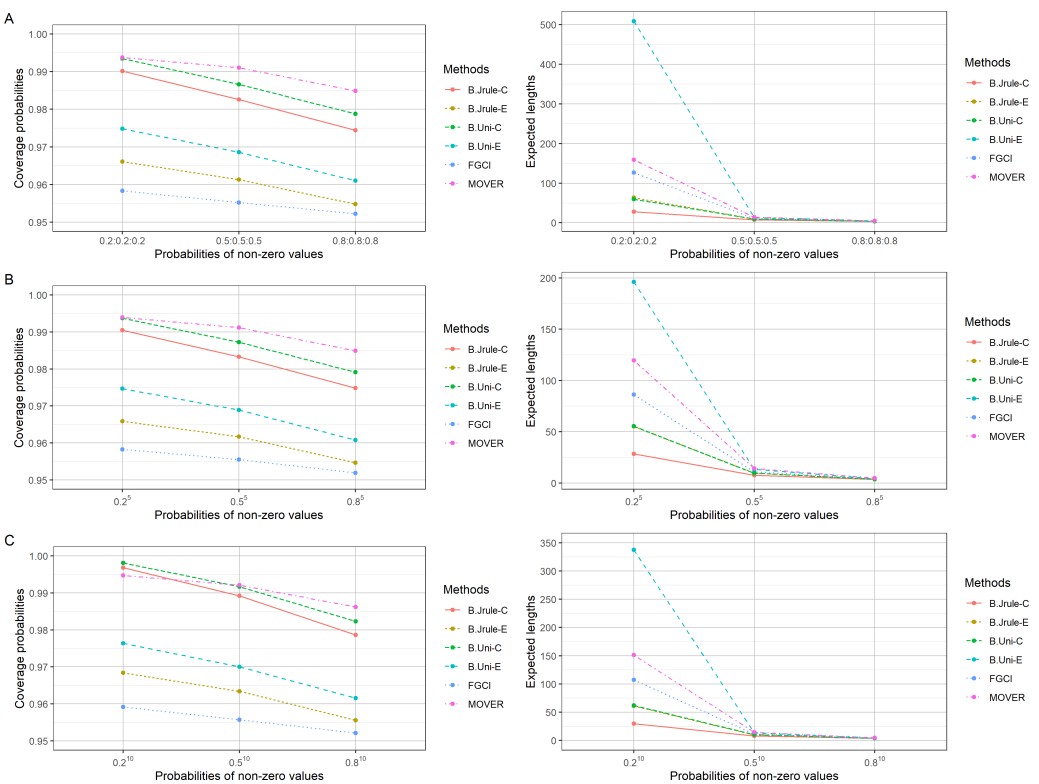

**Figure 2  Comparison of the performances of the proposed methods in terms of their coverage probabilities and expected lengths with various probabilities of non-zero values: (A) $k = 3$ (B) $k = 5$ (C) $k = 10$.**

with *Yosboonruang, Niwitpong & Niwitpong (2020)* who found that the highest posterior density Bayesian using Jeffreys' rule prior is appropriate for constructing the confidence interval for the difference between two independent CVs of delta-lognormal distributions. However, *Abdel-Karim (2015)* and *Thangjai, Niwitpong & Niwitpong (2019)* reported that MOVER is the most suitable for constructing SCIs for the mean or CV of a lognormal distribution, but this is not in agreement with our findings for the data and scenario used in this study since the range of intervals for its SCI was wider than when using the Bayesian methods. In addition, the SCI range between the CVs of the daily rainfall data series from the five different areas of Thailand was too wide, and so this demonstrates that it is different in rainfall dispersion from five areas in Thailand.

## CONCLUSIONS

Herein, we proposed methods to construct the SCIs for all pairwise differences between the CVs of delta-lognormal distributions, including FGCI, two Bayesian methods constructed under the equal-tailed confidence intervals and credible intervals using the Jeffreys' rule and uniform priors, and MOVER. The performances of the proposed methods were determined via their coverage probabilities together with their expected lengths under

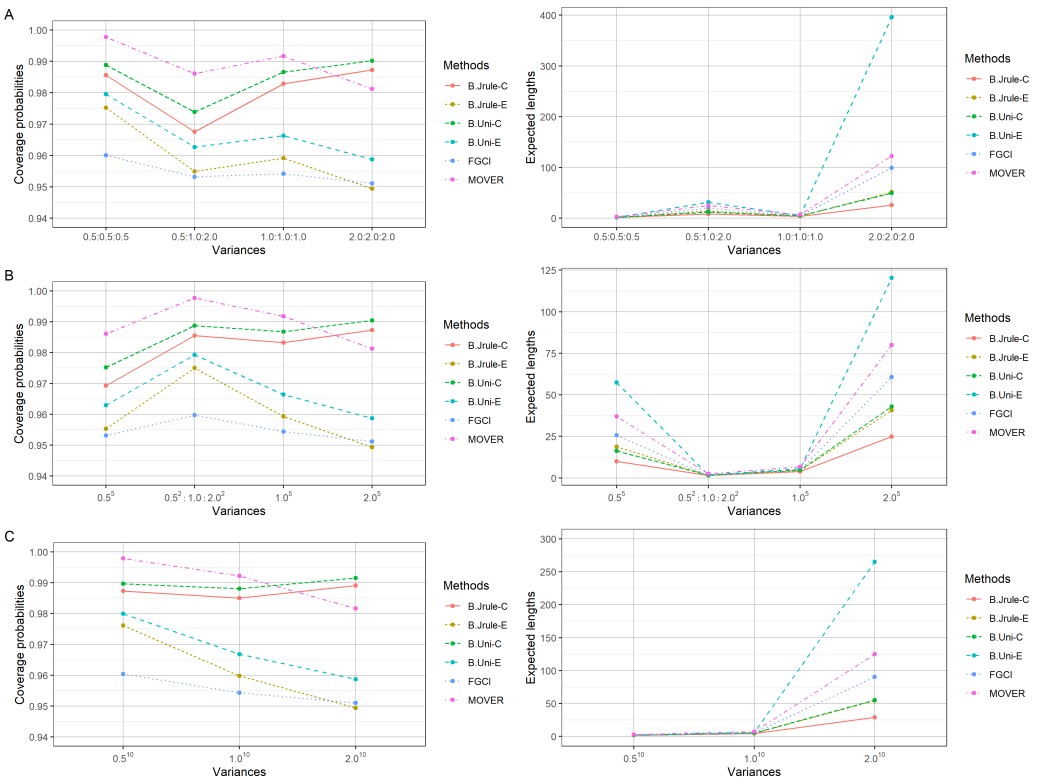

**Figure 3  Comparison of the performances of the proposed methods in terms of their coverage probabilities and expected lengths with various variances: (A) $k = 3$ (B) $k = 5$ (C) $k = 10$.**

**Table 5  AIC and BIC results for testing the distributions of the positive daily rainfall data from the five areas of Thailand in August 2020.**

| Regions | AIC | | | | BIC | | | |
|---|---|---|---|---|---|---|---|---|
| | Normal | Lognormal | Cauchy | Exponential | Normal | Lognormal | Cauchy | Exponential |
| Northern | 200.1677 | 126.6431 | 154.5509 | 143.7143 | 202.3498 | 128.8252 | 156.7329 | 144.7143 |
| Northeastern | 186.9685 | 145.5208 | 170.8114 | 151.8661 | 189.0576 | 147.6098 | 172.9005 | 152.9106 |
| Central | 187.4002 | 142.8220 | 159.7491 | 148.7169 | 189.3916 | 144.8135 | 161.7405 | 149.7126 |
| Eastern | 129.2900 | 99.5405 | 110.7899 | 100.7356 | 130.7061 | 100.9566 | 112.2060 | 101.4437 |
| Southern | 140.0174 | 111.4550 | 124.1323 | 114.2896 | 141.4335 | 112.8711 | 125.5484 | 114.9977 |

various circumstances. The results indicate that the Bayesian credible interval using the Jeffreys' rule prior was suitable for constructing the SCIs for all pairwise differences between the CVs of delta-lognormal distributions in terms of the coverage probability together with the expected length. Furthermore, FGCI is appropriate for constructing these SCIs in cases of the variances equal to 0.5 and 1.0 with the proportion of non-zero values equal to 0.5 and 0.8 for the sample sizes of 50 and 100. In addition, the results of using daily rainfall data from five regions in Thailand coincided with those from the simulation study.

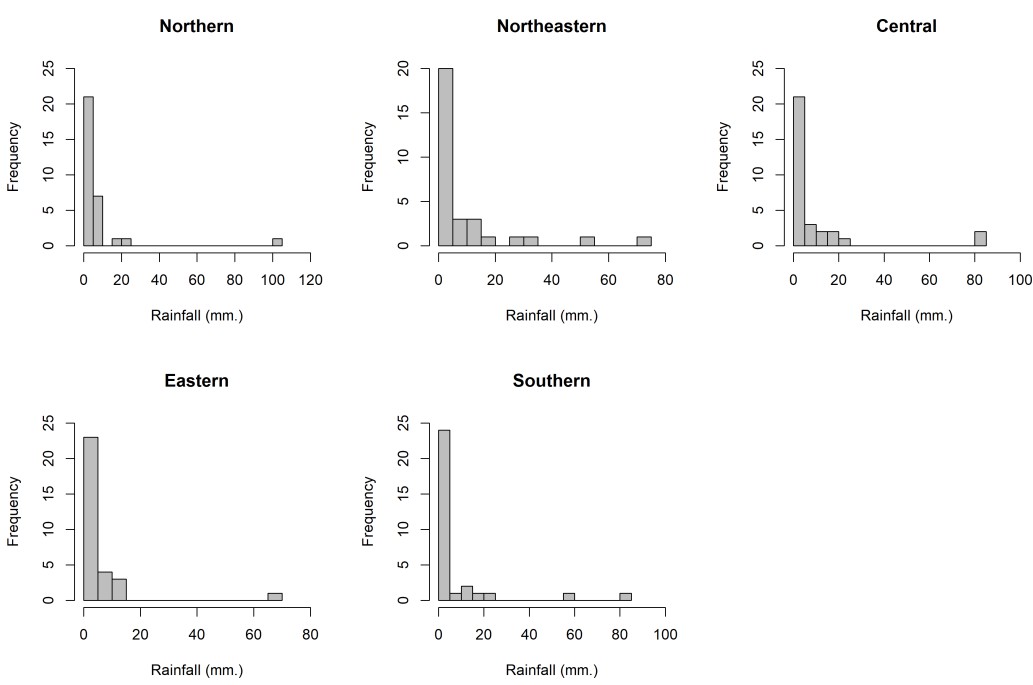

**Figure 4** The density of daily rainfall data in the five areas of Thailand in August 2020.

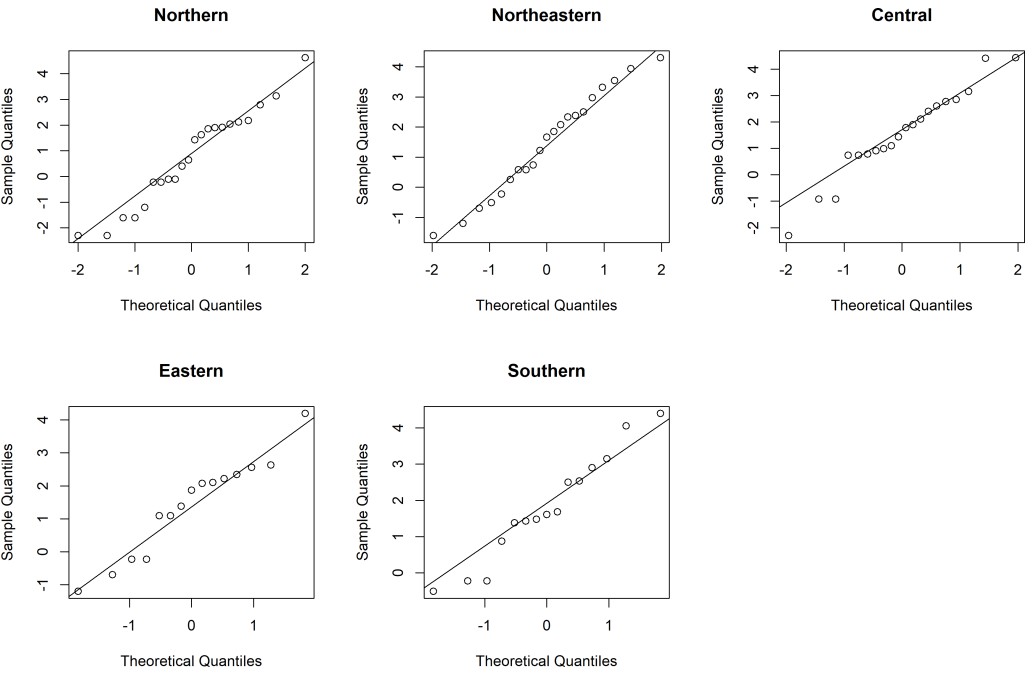

**Figure 5** Normal Q-Q plots of the log-transformed positive daily rainfall data from the five areas of Thailand in August 2020.

**Table 6** The 95% two-sided confidence intervals and credible intervals for all pairwise differences between the CVs of daily rainfall data from the five areas of Thailand in August 2020.

| Regions | $CI_{FGCI}$ | $CI_{B.Jrule-E}$ | $CI_{B.Uni-E}$ | $CI_{B.Jrule-C}$ | $CI_{B.Uni-C}$ | $CI_{MOVER}$ |
|---|---|---|---|---|---|---|
| A1-A2 | [−18.6001,34.6174] | [−18.9230,29.7356] | [−17.4271,40.6785] | [−20.0641,27.7802] | [−22.5574,31.0130] | [−24.4099,41.5271] |
| A1-A3 | [−17.9489,34.8373] | [−14.7811,30.1761] | [−19.4179,40.0759] | [−18.8783,24.1259] | [−20.8181,36.4352] | [−24.6036,41.6372] |
| A1-A4 | [−12.0747,36.3137] | [−11.4058,31.0863] | [−13.2535,43.2976] | [−15.6054,24.9219] | [−18.9824,33.3564] | [−19.2874,42.6266] |
| A1-A5 | [−14.6835,36.5005] | [−12.4631,30.6305] | [−15.3529,42.0874] | [−15.9005,25.3713] | [−18.3991,35.6191] | [−20.8273,42.5263] |
| A2-A3 | [−19.1684,19.8007] | [−15.4068,20.8296] | [−21.9396,19.5370] | [−16.9424,18.6778] | [−18.6853,21.7661] | [−25.8643,25.7772] |
| A2-A4 | [−14.2284,20.9574] | [−12.1077,21.4176] | [−15.2508,20.4711] | [−12.3215,21.1200] | [−16.0818,19.4744] | [−20.5246,26.7455] |
| A2-A5 | [−15.1643,20.7719] | [−14.3124,21.0636] | [−16.9309,20.4638] | [−15.1223,19.4770] | [−15.7284,20.9116] | [−22.0733,26.6474] |
| A3-A4 | [−14.0533,20.3410] | [−12.8066,16.4067] | [−16.1177,22.6669] | [−13.4979,15.6852] | [−17.3305,21.2102] | [−20.6315,26.9413] |
| A3-A5 | [−15.2305,20.4287] | [−14.3490,16.8954] | [−17.6976,22.7643] | [−14.9963,16.0516] | [−16.0804,23.8267] | [−22.1806,26.8432] |
| A4-A5 | [−17.1536,14.8658] | [−15.0784,13.8954] | [−18.1792,16.4151] | [−14.9530,14.0968] | [−18.1763,16.4173] | [−23.1424,21.4920] |

### Funding

This research was funding by King Mongkut's University of Technology North Bangkok. Grant number: KMUTNB-BasicR-64-32. The funders had no role in study design, data collection and analysis, decision to publish, or preparation of the manuscript.

### Grant Disclosures

The following grant information was disclosed by the authors:
King Mongkut's University of Technology North Bangkok: KMUTNB-BasicR-64-32.

### Competing Interests

The authors declare there are no competing interests.

### Author Contributions

- Noppadon Yosboonruang and Suparat Niwitpong conceived and designed the experiments, performed the experiments, analyzed the data, prepared figures and/or tables, authored or reviewed drafts of the paper, and approved the final draft.
- Sa-Aat Niwitpong performed the experiments, analyzed the data, prepared figures and/or tables, authored or reviewed drafts of the paper, and approved the final draft.

### Data Availability

All the data and R code are available as Supplemental File.
The data from northern, northeastern, and southern regions of Thailand were collected by the Northern, Upper Northeastern, and Southern East Coast Meteorological Centers and published at:
- Northern Meteorological Center: Available at http://www.cmmet.tmd.go.th/,
- Upper Northeastern Meteorological Center: http://www.khonkaen.tmd.go.th/Home.php,

- Southern-East Coast Meteorological Center: http://www.songkhla.tmd.go.th/.

The data from central Thailand was collected by the Thai Meteorological Department at: https://www.tmd.go.th/en/.

The data from eastern Thailand was obtained from the Eastern Region Irrigation Hydrology Center at: http://hydro-6.com/.

## Supplemental Information

Supplemental information for this article can be found online at http://dx.doi.org/10.7717/peerj.11651#supplemental-information.

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
