# Peer review of "Simultaneous confidence intervals for all pairwise differences between the coefficients of variation of rainfall series in Thailand"

_PeerJ, doi:10.7717/peerj.11651_

## Round 0.1 · original submission · Major Revisions

The paper has to be deeply revised. The reason for selecting the three methodologies and the place to be studied should be included in the manuscript. The real aim of the work has to be better explained at the end of the introduction section and a real discussion (almost missing in the present paper) of the obtained data has to be added.

·

Basic reporting

This paper is about modelling distributions with a spike at zero which is an important issue. I tis sugessted to model this by means of log-normal distribution for positive rainfall data ann a one mass distribution at zero when there is no rainfall. Quite appropriate. then the focus is on taking all pariwise differences between coefficinet of variaitons in a set of subgroups and the authors develop some methodology for creating simulatanuous 95% confidenc eintervsl. The paper is well written and as all aspects of interest covered.

Experimental design

not applicable as data were routinely collected.

Validity of the findings

Seesm fine to me.

Additional comments

Make sure that in all graphs showing coverage probability the baseline 0.95 is visible. This allows better assessemnt whether the level is kept or not. Please also make the lines bit thicker in the graphs.

Fine paper, my congralustions.

Reviewer 2 ·

Basic reporting

The study proposes the use of three different methods to construct simultaneous confidence intervals (SCI) for all pairwise differences between the coefficients of variation of delta-lognormal distributions of rainfall time series to measure the dispersion of rainfall in five different geographic regions in Thailand. I found some merits in both the methodology and results. In my opinion, this paper has a good potential to be published in the journal. However, I have also some concerns on the different parts of the manuscript.
1: The authors need to clearly define the novelty of the paper. I find it difficult to understand if the authors are just using these different methods to demonstrate the efficacies of the proposed methods to identify the dispersion in rainfall time series or if they are focused on identifying the dispersion in rainfall time series in Thailand for some particular reason. One question is there any significance for choosing the particular study area. I feel that the authors need to explain the scope for conducting this study in the particular study area and support its significance with literature.
2: Line: 36-104 Although the authors has provided the findings of various studies in the literature especially the use of CV and various distributions and methods to investigate the dispersion the authors has failed to describe why they have selected ‘delta-lognormal distribution’ or why they have used the specific methods e.g. fiducial generalized confidence interval in this particular study. The authors should dedicate a paragraph to explain their reasoning behind selecting their methodology.

Experimental design

Line: 228-229 In Table 3 We can certainly see that that there is over-estimation based on coverage probabilities with Jeffrey’s rule from the monte-carlo simulation study. The authors however blindly accept to measure the performance purely based on the expected lengths (shorter the better) and claim the particular method is better. Why is FGCI which has close expected lengths but better coverage probabilities could be the right method for SCI estimation in the study area?

Line: 248-249 ‘FGCI is appropriate for constructing these SCIs in cases of small variance with a 249 medium-to-high proportion of non-zero values and a medium-to-large sample size’ this particular conclusion is not supported in the main body of the manuscript. The authors should clearly state the conclusion with focus on the study area since there no conceivable conclusion presented to show any progressive findings discovered in this study.

Validity of the findings

In the discussion section, the authors should discuss the physical meaning of the findings e.g. whether the SCI range between the CVs of daily rainfall data in the five different areas of Thailand is good or bad. The author should calculate the SCI for the five areas separately and provide a separate table rather than combining all the areas into a single combined area (Table 5), which might create a significant statistical drift in the results.

The authors should add additional references in the discussion section and compare the findings with previous studies and establish the significance of the particular study in Thailand.

Line:19 In Abstract, the authors use the term “coverage probabilities”. Explain the meaning of the term. Also explain the need for estimating coverage probabilities in the Introduction section.

Additional comments

The authors should add a map/Table to show how they have segregated the five areas for study in Thailand (and explain if there is any reasoning behind such classification in the introduction section).

Abstract can be improved especially in defining the study’s problem statement.

The language used throughout the manuscript is adequate, however, there is scope for improvement notably in the introduction section.

Reviewer 3 ·

Basic reporting

This paper construct simultaneous confidence intervals for all pairwise differences between the coefficients of variation of delta-lognormal distributions using three methods: fiducial generalized confidence interval, Bayesian, and the method of variance estimates recovery.

The paper was be written in professional English. The paper structure, figures and tables is suitable.

Experimental design

There are some points that the authors should correct and improve as follows:
1. The authors should provide the difference between confidence interval and credible interval.
2. In Table 5, the interpretation of the SCIs and credible intervals for all pairwise differences between the CVs must be conducted.
3. In practice, how do the researcher select the Bayesian priors (The Jeffreys’ rule prior and Uniform prior). The author should give the criteria for selection.

Validity of the findings

This paper proposed the new knowledges in statistical theory and applied them in the real situation.

Annotated reviews are not available for download in order to protect the identity of reviewers who chose to remain anonymous.

---

## Round 0.2 · accepted · Accept

I am pleased to confirm that your paper has been accepted for publication in PeerJ.

Thank you for submitting your work to this journal.

·

Basic reporting

no further comments.

Experimental design

ok

Validity of the findings

ok

Additional comments

no further comments

Reviewer 2 ·

Basic reporting

The revised article meet the standards

Experimental design

The revised article is now complete and meet the standards

Validity of the findings

The results of revised article are valid

Additional comments

Congratulations for your efforts

Reviewer 3 ·

Basic reporting

This paper construct simultaneous confidence intervals for all pairwise differences between the coefficients of variation of delta-lognormal distributions using three methods: fiducial generalized confidence interval, Bayesian, and the method of variance estimates recovery.

This paper was written clear and has professional English language usage.

Experimental design

There are few issues for improving as follows:

1. The authors should mention the difference between confidence interval and credible interval.
2. In Algorithms 1 and 2, repeat Steps 2-3 5,000 times and repeat Steps 1-5 15,000 times. How the authors known the number of the repeat? The author should explain the reason.
3. The 95% two-sided confidence intervals and credible intervals must be interpreted.
4. The future research must be added in the paper.

Validity of the findings

This paper proposed the new knowledge in statistical theory and applied them in the real situation.

Additional comments

In my opinion, this paper is suitable and can be published in PeerJ journal.